# Can Rural Road Construction Promote the Sustainable Development of Regional Agriculture in China?

**Zhou Zhou, Jianqiang Duan \*, Wenxing Li and Shaoqing Geng** 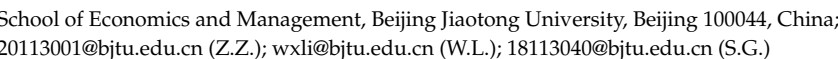

School of Economics and Management, Beijing Jiaotong University, Beijing 100044, China; 20113001@bjtu.edu.cn (Z.Z.); wxli@bjtu.edu.cn (W.L.); 18113040@bjtu.edu.cn (S.G.)

\* Correspondence: jqduan@bjtu.edu.cn

**Abstract:** The sustainable development of agriculture is significant in protecting natural resources, protecting the ecological environment, ensuring food security, and eliminating poverty. Rural road construction promotes the flow of labor and capital between urban and rural areas, and plays a vital role in agricultural production and rural revitalization. This study aims to analyze the effect of rural road construction on the sustainable development of regional agriculture in China. We select five-dimensional indexes of population, society, economy, resources, and environment and use the entropy method to calculate the agricultural sustainable development index of each province in China. Then, we construct the spatial econometric model to explore the influence based on the panel data of 31 provinces in China from 2002 to 2018. The benchmark results show that rural road construction significantly promotes the sustainable development of agriculture; however, it has a negative impact on environmental sustainability, and the influence is lagging. The results are also heterogeneous among regions. Rural road construction has significantly promoted the sustainable development of agriculture in eastern and central areas, but has no significant impact on western regions. The reason is that the siphoning effect caused by the construction of rural roads has led to a loss of talents and capital in the western region, which harms the sustainable development of the population and resource system. This effect offsets the positive effect of the other three systems. This research has substantial policy implications for promoting rural revitalization and agricultural development.

**Keywords:** rural roads; sustainable agriculture development; entropy method; spatial econometric analysis

## 1. Introduction

Agriculture is the foundation of national economic development and occupies an important position in social development. Its long-term stable and coordinated development directly affect national economic growth and social stability [1,2]. The sustainable development of agriculture has crucial scientific value in protecting natural resources, maintaining the ecological environment, ensuring food security, eliminating poverty, and even promoting rural revitalization [3–5]. Recently, the agricultural economy has developed rapidly because of science and technology and the accelerated process of agricultural modernization. However, it has brought about many problems, such as the over-exploitation of farming resources, the excessive use of agricultural inputs, the over-extraction of groundwater, a large degree of social resource consumption, environmental pollution, and other issues [6]. These have challenged the sustainable development of agriculture to some extent. This issue is of great concern to the country. To promote the sustainable development of agriculture in China, the Chinese government is now deeply aware of the importance of sustainable agricultural development, and has issued a series of documents. Therefore, exploring the constraints of sustainable agricultural development has important practical significance. Consequently, it is essential to explore the factors that affect the sustainable development of agriculture in China.

Generally speaking, vigorously developing transportation infrastructure is an important measure to promote rural revitalization and agricultural production. Improving transportation infrastructure can encourage agricultural labor productivity by enabling labor transfer to the non-agricultural sector, which could help labor transferred to cities earn more money in the city. Meanwhile, it could also be conducive to improve agricultural mechanization [7–10]. However, related studies have shown that road construction does not always promote local economic development [11]. It imposes more extensive restrictions on the environmental goals of sustainable development in rural areas, especially the rational use of land [12]. On the one hand, rural roads incur enormous ecological and social costs during the construction and operation period [13]. On the other hand, rural road construction will also cause problems such as land fragmentation and deterioration of arable land layout [14]. These will severely restrict the sustainable development of agriculture.

Rural roads are a vital part of the road network. They play a role in connecting highways and arterial roads with towns, villages, and tourist spots. They directly serve farmers and the rural economy. It is an important carrier to communicate the flow of people, logistics, and information in rural areas. Since the 21st century, to build a moderately prosperous society in an all-around way, China's government has paid more attention to rural roads, significantly increasing investment in rural road construction and continuously improving the layout of rural road construction. As of the end of 2020, China's rural road mileage was 4.3823 million kilometers, which is an increase of 181,800 km over the end of the previous year, accounting for 84.3% of the total road mileage, which is a rise of 0.5%. Because of the importance of rural roads to the rural economy and the rapid development of rural roads in China in recent years, this article intends to explore the impact of rural road construction in China on the sustainable development of agriculture. The discussion of this issue can provide specific experience support for the realization of rural revitalization strategies.

Investment in transportation infrastructure is the advanced capital, which exerts a substantial economic effect. Moreover, the "economic growth effect of transportation infrastructure" has been focused on by academic circles. Generally speaking, transportation infrastructure construction is considered an essential factor of economic growth [15–17]. On the one hand, a significant infrastructure investment produces a multiplier effect, which drives regional economic growth [18–20]. On the other hand, transportation infrastructure can promote the flow of production factors, optimize the allocation of regional resources, improve the operational efficiency of enterprises, and ultimately promote economic growth [21–24]. Although many studies have analyzed the impact of transportation infrastructure on social and economic activities, most of the research objects are industrial sectors or regional economic development. Few pieces of research have explored the impact of transportation infrastructure on rural area development and agricultural production.

The existing literature on transportation infrastructure, agriculture, and rural areas can be divided into three categories.

The first category is transportation infrastructure and the urban–rural income gap. Liu [25] found that improving highway accessibility in China significantly increased residents' income, and rural areas benefited more than urban areas. Yang [26] found that railway speed-up can narrow the income gap between urban and rural areas by promoting the flow of labor, capital, and technology. Chen [27], Yu [28], and Li [29] all found that the development of a high-speed rail significantly narrowed the income gap between urban and rural areas. However, Yu [30] found that the income gap between urban and rural areas in cities with high-speed rail is aggravated compared with cities without high-speed rail. This phenomenon is particularly significant in the eastern region, Beijing–Tianjin–Hebei region, Yangtze River Delta, and Pearl River Delta economic circle.

The second category is transportation infrastructure and agricultural productivity. Aggarwal [31] found that rural road construction enhances rural connectivity, promotes rural families to increase the use of rural technology, and improves labor productivity.

Stoelo [32] built a heterogeneous agricultural model to explore the relationship between trade, productivity, and welfare. They found that road construction will improve the overall efficiency by 4.9%. Alberto [33] found that reducing trade costs will lead to many labor transfers from food to cash crops. At the same time, it will also increase investment in modern agriculture, thus improving labor productivity. Teng and Li found that highway construction can boost agricultural productivity. There are three main mechanisms: agricultural mechanization, non-agricultural labor transfer, and market access [9,34].

The third category is transportation infrastructure and agricultural land, and regional sustainable development. The construction of highways has brought many negative impacts, especially in the farmland area that the highway passes through [35], which will have a greater impact on the spatial structure of the land, such as land fragmentation [36,37]. Dispersing the land will incur additional costs and the time expenditure for obtaining land [38], which will increase the cost of agricultural production to a certain extent and reduce the efficiency of the agricultural output. At the same time, highway construction will also have a more significant negative impact on landscape structure, urban land cover, biodiversity, and landscape diversity, thereby restricting the region's sustainable development [39].

Existing literature has fully explored the impact of transportation infrastructure construction on economic growth, agricultural land, and the environment. However, these all discuss the impact of transportation infrastructure construction from a single dimension, and have not comprehensively considered its effect on the sustainable development of agriculture. This paper plans to select five-dimensional indicators of population, society, economy, resources, and environment to calculate the regional agricultural sustainable development index. Then, we construct the spatial econometric model to explore the impact of rural road construction on the sustainable development level of agriculture in China. We have obtained the following research conclusions: (1) On the whole, rural road construction significantly promotes agriculture's sustainable development. Still, it has a negative impact on the sustainable development of the environment. (2) From a regional perspective, the construction of rural roads has significantly promoted the sustainable development of agriculture in the eastern and central regions. However, its impact on the sustainable economic development of the two regions is not significant, and it also has a substantial negative impact on the environmental sustainability of the central region. For the western region, the impact of rural road construction is not significant. The reason is that the siphoning effect brought by the rural road construction has caused a considerable loss of talents and capital in the western region, which in turn caused the sustainability of its population and resource system to decline. The positive effects of society, economy, and environment are offset. The research conclusions have substantial policy implications for promoting rural revitalization and agricultural development in China. For example, the eastern and central regions should strengthen the guidance of local farmers in the scientific application of chemical fertilizers and pesticides, increase the application of organic fertilizers, and implement straw return to the field to reduce the impact of chemical fertilizers and pesticide pollution on agricultural production. The western region needs to give some preferential policies, such as talent subsidies, agricultural subsidies, etc., to attract talents and capital inflow, promote the accumulation of local human capital, and improve the level of agricultural mechanization to enhance the agricultural labor productivity in the western region.

The research contribution of this paper is mainly manifested in the following three aspects.

Firstly, although some kinds of literature have recently explored the relationship between road construction and agricultural productivity, the study of its effect on the sustainable development of agriculture is scarce. This paper enriches and develops this kind of literature.

Secondly, the available literature has explored the impact of large-scale transportation infrastructure construction, such as highway construction and high-speed rail construction,

on agricultural development. However, these researches have ignored the impact of rural roads, which directly connect urban and rural areas. This paper plans to explore the relationship between rural road construction and sustainable agricultural development.

Thirdly, the existing literature rarely analyzes the inter-regional correlation from the perspective of spatial heterogeneity. However, rural road construction promotes inter-regional correlation by reducing transportation costs. Therefore, ignoring spatial correlation will lead to biased regression results. Thus, this paper constructs a spatial econometric model, which adds spatial lag variables to the benchmark model to explore the spatial correlation in the sustainable development of regional agriculture.

The remainder of this study is structured as follows. Section 2 introduces the construction of rural roads in China. Section 3 constructs a theoretical framework to explore the relationship between rural road construction and regional agricultural sustainable development. Section 4 introduces the process of building the model and selecting the corresponding indexes. In Section 5, we mainly present regression results and the robustness test. We also do the heterogeneity analysis in different periods and regions in Section 6. The last section is the conclusion and policy implications.

## 2. Introduction of Rural Road Construction in China

### 2.1. Rural Road Development in China

Transportation is a fundamental and leading industry in the national economy and an important service industry. Since 2010, the total mileage of highways in China has continued to increase. In 2019, China's total highway mileage was 5.0125 million kilometers, increasing 166,000 km over the previous year. In particular, China's rural roads have developed rapidly in recent years. As it is shown in Figure 1, from 2015 to 2019, the overall mileage of rural road construction in China has shown a general growth trend. At the end of 2019, China's rural road mileage reached 4.205 million kilometers, which is an increase of 4.0% over 2018. In recent years, the growth rate of the rural road mileage exceeded that of the total road construction mileage for the first time, and was 0.6 percentage points higher. As it is shown in Figure 2, from 2015 to 2018, China's rural road construction investment has increased year by year, and the growth rate of rural road investment is higher than that of total road construction investment. Although China's rural road construction investment had slowed down significantly in 2018, the investment efficiency of China's rural road construction is improving because of the year-on-year increase in the mileage of China's rural roads.

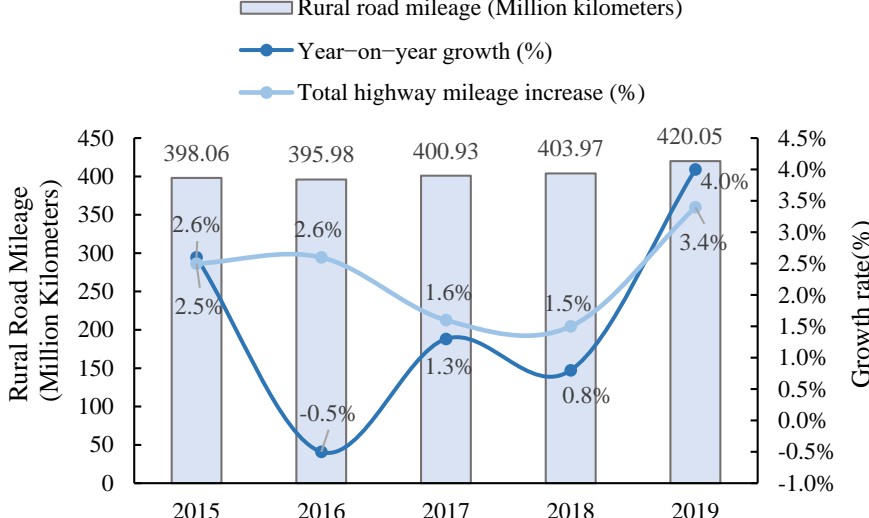

**Figure 1.** Mileage and growth of rural road construction in China from 2015 to 2019 (10,000 km, %). Data source: Ministry of Transport of the People's Republic of China.

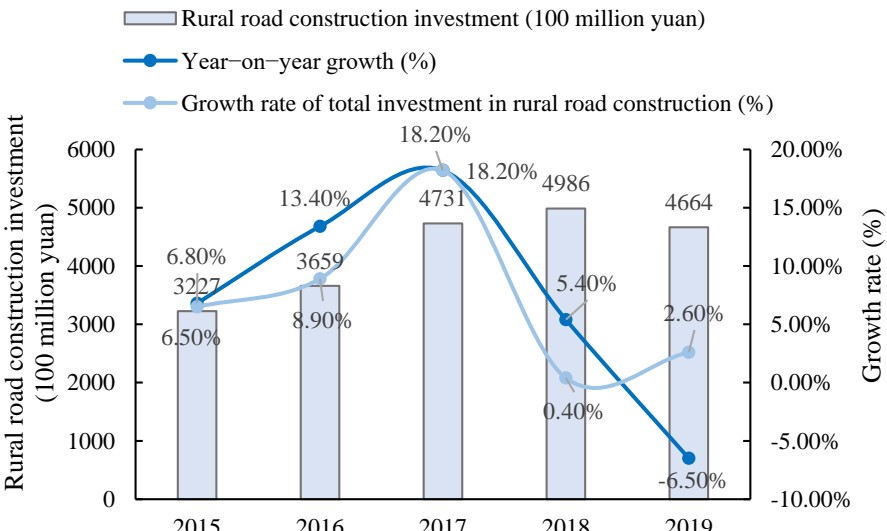

**Figure 2.** Rural road construction investment and growth in China from 2015 to 2019 (100 million yuan, %). Data source: Ministry of Transport of the People's Republic of China.

### 2.2. Introduction to China's Rural Road Construction Related Policies

At present, China's rural road construction has achieved excellent results under the promotion of national policies. It has proposed a clear development direction for the management level of rural road construction and network layout, effectively promoting the rural economy around the road. The relevant policies in recent years are shown in Table 1.

**Table 1.** China's rural road construction-related policies.

| Year | Policy | Main Content |
|------|--------|--------------|
| 2018 | Measures for the Quality Management of Rural Road Construction | Clarify rural road construction projects, implement a life-long quality responsibility system, provide a basis for rural road construction, and make rural road construction more standardized with laws to follow. |
| 2019 | Guiding Opinions on Promoting the High-quality Development of the "Four Good Rural Roads" | It is required that the construction of rural roads be adapted to local conditions and be people-oriented, compatible with the optimization of the layout of villages and towns, the development of rural economy, and the safe and convenient travel of farmers. It is necessary to build, manage, protect and operate rural roads further and gradually eliminate the traffic that restricts rural development. The bottleneck provides a better guarantee for the majority of farmers to shake off poverty and become prosperous. |
| 2021 | Outline for the Medium and Long-term Development of Rural Highways | By 2035, a rural road transportation system with "reasonable scale and structure, high-quality facilities, standardized and effective governance, and high-quality transportation services" will be formed. The total mileage of rural roads will exceed 5 million kilometers. |

### 2.3. Introduction to China's Graded Highways

According to China's "Highway Engineering Technical Standards" (JTG B01-2014) on the classification standards of graded roads, the roads are divided into highways, first-class roads, second-class roads, third-class roads, and fourth-class roads to their tasks, functions, and flows. It is thus made up of five levels. Among them, highways and first-class roads are high-grade highways, while second-class, third-class, and fourth-class roads are considered low-grade (see Table 2 for details).

| Category | Functions | Explanations |
|---|---|---|
| Highway | Mainly used to connect politically, economically, and culturally significant cities and regions | Connecting cities and cities |
| First-class | Mainly connect important political and economic centers and lead to crucial mining areas | |
| Second-class | Arterial roads connecting political and economic centers, large industrial and mining areas, or suburban roads with heavy traffic | Connecting the city and the countryside |
| Third-class | A general arterial road connecting county and above towns | |
| Fourth-class | Branch roads connect counties, townships, villages, etc. | |

Resources source: China's "Highway Engineering Technical Standards" (JTG B01−2014).

As seen from the above table, China's highways and first-class highways mainly connect core cities and directly impact the rural economy. Although the secondary road mainly connects significant cities, it also connects the suburbs, indirectly affecting the rural economy. Third-class and fourth-class roads directly connect towns and rural areas. Thus, we mainly focus on the impact of second-, third-, and fourth-class roads on agricultural development.

## 3. Theoretical Analysis

Sustainable development mainly includes the degree of the economy, society, ecology, resources, etc. It is a new development thinking pattern of humanity [40]. The sustainable development of agriculture is an essential part of realizing sustainable development, which has attracted much attention from academic circles [41]. When The Food and Agriculture Organization (FAO) convened the International Conference on Agriculture and Environment in the Netherlands, it extended the meaning of sustainable agricultural development to sustainable agriculture and rural development, with the following three goals: the first is to alleviate poverty and realize comprehensive development in rural areas; the second is to make sure the food production could increase continually and safety; and the third is to complete a virtuous circle of the environment and resource protection.

Based on the ideas and goals of sustainable development, sustainable agriculture could be concluded the following characteristics:

- Population sustainability. This is an essential requirement of sustainable development. The continuous improvement of rural residents' quality is based on controlling population growth and appropriately transferring surplus rural labor to cities. Population sustainability mainly focuses on whether the number of residents is sustainable.
- Social sustainability. This is the fundamental goal of sustainable development. It includes embodying the principle of fairness in using social-economic and rural natural resources so that rural residents' living standards and quality will continue to improve the second bullet. Social sustainability mainly focuses on whether the residents' life quality is sustainable.
- Economic sustainability. This includes enabling agricultural production that is profitable, and maintaining a high level of profitability for a long time.
- Resource sustainability. This emphasizes the sustainable use of natural resources necessary for agricultural production and adopting various measures to protect natural resources, including the sustainable use of water resources, stabilizing and increasing the area of arable land, etc.
- Environmental sustainability. This refers to the excellent maintenance and improvement of the ecological environment that affects and restricts agricultural production, including the maintenance of water resources, atmosphere, soil, and other agricultural ecological environments.

In summary, to meet the sustainable development of agriculture, it is necessary to ensure the mobility of the population and resources. Transportation has always been considered an essential factor in social change [42]. The construction of rural roads reduces the transportation cost of human, capital, and goods between regions and urban and rural areas. It also improves transportation efficiency, which directly or indirectly affects the sustainable development of agriculture.

Based on the above sorting and analysis, the research framework designed in this paper is shown in Figure 3.

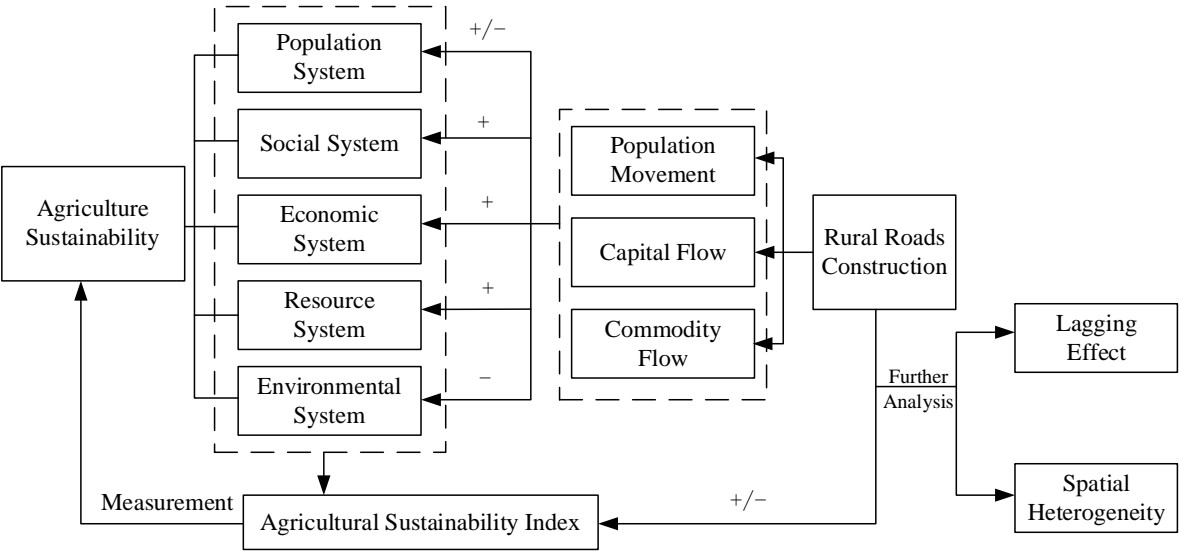

**Figure 3.** Logical frame diagram of the relationship between rural road construction and sustainable agricultural development.

In the above figure, the construction of rural roads has promoted or inhibited the development of the regional agricultural population, society, economy, resources, and environmental systems by improving the flow efficiency of population, capital, and commodities, which in turn affected the availability of the agricultural sustained development ability. This impact is reflected in the overall effect of rural road construction on sustainable agricultural development, the various subsystems of regional agricultural sustainable development, the lagging impact in time, and the heterogeneous impact in space.

Based on the above research framework, this paper makes a qualitative analysis of various influences and puts forward relevant hypotheses.

### 3.1. Rural Road Construction and Sustainable Agricultural Development

Rural road construction improves the accessibility of urban and rural areas, and reduces transaction costs in the flow of urban and rural elements. Reduced transportation costs help to further strengthen the flow of resources or elements between urban and rural areas, thereby triggering the reintegration of urban and rural element resources. The redistribution effect of urban and rural factors caused by the construction of rural roads will have a dual impact on the sustainable development of agriculture.

On the one hand, the construction of rural roads has strengthened the economic ties between urban and rural areas and significantly weakened the time and space barriers in the flow of urban and rural elements [26]. The superior employment, housing, education environment, and agglomeration effects in urban areas have led to the flow of many agricultural factors and resources into urban areas. Although this has promoted the rapid development of urban areas, it will also bring about a considerable loss of production factors in the agriculture or rural sectors [14], breaking the balance of the rural ecological environment [35–39], thereby inhibiting the sustainable development of agriculture.

On the other hand, the construction of rural roads helps transfer surplus rural labor to cities and towns, creates conditions for the transfer of rural land, and helps to promote

the development of agricultural specialization [10]. At the same time, the construction of rural roads enhances the inter-industry linkage effect, helps to strengthen the secondary and tertiary industry's "technological diffusion effect" on the agricultural sector. Improved efficiency of goods or services supply will increase agricultural labor productivity and land output rate [9]. In addition, the reduction in transportation costs brought about by the construction of rural roads is also conducive to the inflow of advanced production factors such as high-quality urban talents, technical management, and market concepts to agriculture. From the above analysis, the construction of rural roads has dramatically promoted increased agricultural productivity, which enabled the sustainable development of agriculture.

Therefore, the impact of rural road construction on the sustainable development of agriculture depends on the comprehensive performance of inhibition and promotion.

**Hypothesis 1 (H1).** *When the restraint effect of rural road construction on the sustainable development of agriculture is greater than the promotion effect, it is not conducive to the sustainable development of agriculture, and vice versa.*

*3.2. Rural Road Construction and Population, Society, Economy, Resources, and Environmental Sustainability*

The immediate changes brought by the construction of rural roads include the following three aspects: Firstly, the construction of rural roads has dramatically reduced the travel time of residents and increased residents' willingness to travel [26]. Secondly, the improvement of road quality has made the capital, such as agricultural machinery equipment, seeds, etc., to the countryside is faster [9]. Third, the construction of rural roads increases the goods moving speed between regions, which significantly reduces the transportation cost of agricultural products ascending and industrial products descending [33]. The direct impact of the above changes is to accelerate the flow of factors such as population, capital, and commodities, providing more possibilities for the income growth of various factors in rural areas, thereby impacting sustainable agricultural development.

First of all, the construction of rural roads has promoted the flow of humans, hastened the rural residents to enter the city, and encouraged the urban residents to go to the countryside. These factors have jointly promoted the increase of rural residents' income. For rural residents, the construction of rural roads is conducive to the transfer of surplus labor to cities [10], and it also contributes to the accumulation of human capital [43]. From the perspective of labor mobility, the construction of rural roads reduces the space-time distance between urban and rural areas, eliminates the barriers to rural labor mobility, decreases transfer costs, and promotes the flow of surplus rural labor to urban areas. According to the dual economic theory, there is a surplus labor force with zero marginal labor productivity in the agriculture of developing countries. The marginal efficiency of factors and wages in the modern industrial sector are generally higher than those in the agricultural sector, thus attracting the continuous transfer of rural labor. The decreased labor force in the agricultural sector could improve the marginal labor productivity and increase farmers' income [44]. From the perspective of human capital accumulation, rural road construction provides convenient conditions for urban–rural exchanges. The regional co-urbanization effect is conducive to expanding the scope of urban cultural dissemination, enhancing the accessibility of modern ideas and concepts, and increasing the value of rural residents to education. In this way, the willingness of farmers to invest in human capital can be improved, which is exceptionally beneficial for the improvement of the labor productivity of rural residents [45]. For urban residents, the construction of rural roads has dramatically facilitated urban technical personnel to enter the countryside for guidance, improving farmers' production skills, and increasing production efficiency [26].

**Hypothesis 2 (H2).** *The construction of rural roads has accelerated population flow, promoted the transfer of surplus rural labor to cities, and at the same time, improved its human capital. The sustainability of the population system has been continuously enhanced.*

Second, the construction of rural roads promoted the flow of capital [46]. On the one hand, the continuous improvement of road facilities encourages agricultural machinery and equipment transportation from cities to rural areas, improving agricultural mechanization [9,34]. Agricultural mechanization promotes the advancement of agricultural labor efficiency by reducing production costs and increasing agricultural output. At the same time, it is also conducive to a more efficient use of resources. For example, irrigation infrastructure has effectively increased the irrigation rate and promoted the development of resource-saving agriculture [47]. On the other hand, the agglomeration effects exacted by the construction of rural roads enable many factors to flow into the tertiary industry, such as finance, education, and agricultural technology services, which guarantees the sustainability of agricultural fixed-asset investments [48].

Furthermore, the construction of rural roads has accelerated the flow of materials. Rural road construction can accelerate the flow of agricultural products to urban markets and facilitate the introduction of material capital in rural areas. As for agricultural products flow, most agricultural products are perishable commodities, which have high requirements for the timeliness of transportation. In 2019, the total output of agricultural products in China was 1.98 billion tons, of which the production of fresh farm products exceeded 1.1 billion tons, and the loss of fresh agricultural products that could not be sold in time reached 20–30%, while only about 5% in developed countries [26]. The construction of rural roads saves agricultural products' transportation and storage time, reduces the loss of fresh agricultural products due to time and collisions, and increases farmers' income. At the same time, the decline in transportation costs will also help promote the integration of urban and rural markets, expand agricultural sales channels, reduce transaction costs, and increase rural residents' income.

**Hypothesis 3 (H3).** *The construction of rural roads has accelerated the flow of capital and commodities between urban and rural areas and regions, promoting local agricultural mechanization and increased investment in fixed assets, thereby improving the efficiency of agricultural resource utilization and the living standards of rural residents. The sustainability of social, economic, and resource systems can be improved.*

In addition, the construction of rural roads also contributes to the decline of material capital, such as fertilizers, pesticides, and seeds. Since transportation costs are a more critical part of product price additions, the reduction in logistics costs has led to a corresponding drop in product prices. At present, the literature found that the construction of rural roads has significantly reduced the price of fertilizers [31,33], and the use of chemical fertilizers by rural residents has continued to increase. In addition, the construction of rural roads will also promote the interconnection between urban and rural residents. When rural residents see a considerable income gap with urban residents, there will be a specific psychological gap. In order to increase their income level, rural residents may be more inclined to use fertilizers and pesticides to increase production [49]. Road construction's "price effect" allows rural residents to buy more fertilizers based on the two points above.

Furthermore, due to the increase in urban–rural connections and the comparative psychology of rural residents, they are more inclined to increase the use of chemical fertilizers and pesticides. In the short term, fertilizers will help increase productivity and the income of rural residents. However, in the long run, excessive use of chemical fertilizers and pesticides will accelerate soil acidification, increase heavy metals and toxic elements in the soil, and the resulting soil pollution will severely restrict the sustainable development of China's agriculture.

**Hypothesis 4 (H4).** *The construction of rural roads has brought down the transportation cost of chemical fertilizers and pesticides, which has increased the number of farmers using them. Its excessive use has caused environmental problems such as land acidification and soil erosion, and has thus inhibited the sustainable development of agriculture.*

### 3.3. The Lagging Effect of Rural Road Construction on Sustainable Agricultural Development

The construction of rural roads reduces the transportation costs of people, capital, and goods, and accelerates the flow of factors. However, each individual needs to conduct a cost-benefit analysis before deciding to maximize its utility. For the rural surplus labor, it also needs to be weighed in the process of mobility. On the one hand, rural laborers go to areas with better economic development to get more job opportunities and income [10]. On the other hand, rural laborers must face inevitable migration and psychological costs [50,51]. Farmers engaged in agricultural production need to comprehensively measure the importance of transportation costs in their agricultural production process to decide whether to increase their fixed agricultural asset investment [9]. The above decision-making process will delay the impact of rural road construction on agricultural production. In addition, the construction of rural roads itself produces certain negative externalities to the surrounding environment [39], such as environmental pollution during the construction process [14], reduction of land quality [35], etc. This effect will not have a significant impact immediately, but it will restrict the sustainable development of agriculture in the long run. Therefore, the impact of rural road construction has a certain lag.

**Hypothesis 5 (H5).** *There is a certain lag in the impact of rural road construction on the sustainable development of agriculture.*

### 3.4. The Spatial Heterogeneity of the Impact of Rural Road Construction on Sustainable Agricultural Development

The increasingly perfect rural road construction has dramatically reduced transportation costs and has an essential impact on the degree of agricultural mechanization [9]. Agricultural mechanization is a crucial factor in improving the efficiency of China's agricultural production [52]. However, natural geographical conditions, especially topographical conditions, affect rural road construction and agricultural production. Many studies have found that the plain areas represented by the two extensive plains of north China and northeast China, and the hilly areas and mountainous areas defined by the west, especially the southwestern region, are the two extremes of the development of agricultural mechanization in China [53]. Compared with mountainous and hilly areas, the cost of building rural roads in plain areas is lower, and the effect of improving agricultural mechanization is more significant [9]. The topography of the east, central, and west is quite different in China. Mountains, plateaus, and basins dominate the west, and plains and hills dominate the east and middle. This diverse topography causes the impact of rural road construction on local agriculture to be heterogeneous.

**Hypothesis 6 (H6).** *The impact of rural road construction on the sustainable development of agriculture in different regions is heterogeneous.*

## 4. Methods and Data

### 4.1. Econometric Model

This article aims to explore the impact of rural road construction on the sustainable development of China's agriculture. We first calculate the agricultural sustainability index through the entropy method, and then use the provincial panel data from 2002 to 2018 to construct a fixed-effect model to explore the impact of rural road construction. According to the first law of geography, everything is connected. The closer the distance is, the stronger the connection [54]. Therefore, this paper adds spatial factors to the benchmark model, and empirically explores the relationship between rural road construction and sustainable agriculture development by constructing a spatial panel measurement model. And the whole model constructing procedure is shown in Figure 4.

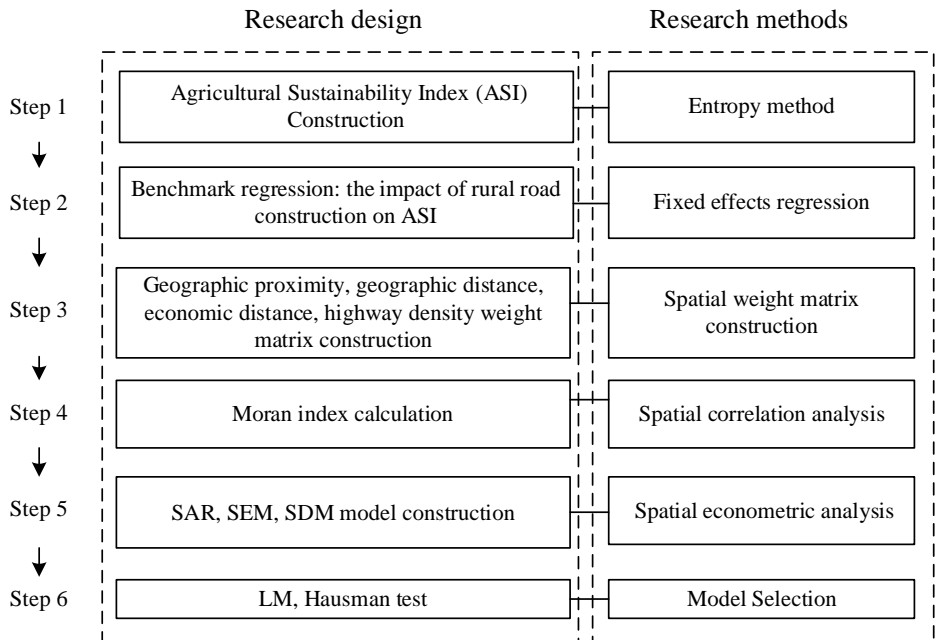

Research design — Research methods

Step 1 — Agricultural Sustainability Index (ASI) Construction — Entropy method

Step 2 — Benchmark regression: the impact of rural road construction on ASI — Fixed effects regression

Step 3 — Geographic proximity, geographic distance, economic distance, highway density weight matrix construction — Spatial weight matrix construction

Step 4 — Moran index calculation — Spatial correlation analysis

Step 5 — SAR, SEM, SDM model construction — Spatial econometric analysis

Step 6 — LM, Hausman test — Model Selection

**Figure 4.** Research design and research methods.

### 4.1.1. Basic Regression: Fixed-Effect Regression

To verify the above assumptions, we build the following econometric model for empirical analysis. First, we take the regional agricultural sustainable development index as the explained variable, rural road construction as the core explanatory variable, and urban–rural income gap, GDP, fiscal expenditure, and other indicators reflecting regional development as the control variables. Then, we use these variables to build a panel model for exploring the relationship between rural road construction and sustainable agriculture development. Meanwhile, rural road construction can promote the interrelation between regions. The rural road development of one area will bring specific spillover effects to surrounding areas, so the spatial correlation between regions could not be ignored. Based on this, this paper adds the spatial lag term of variables to the initial model. According to the above analysis, the initial econometric model took the following form:

$$ASI_{it} = \alpha_0 + \alpha_1 road_{it} + Z_{it}\lambda + \gamma T_t + v_i + u_{it} \tag{1}$$

where the subscript $i$ and $t$ refer to the $i - th$ province and $t - th$ year, respectively. $ASI_{it}$ is the indicator of regional agricultural sustainable development, $road_{it}$ represents the construction of rural roads, and $Z_{it}$ is a series of control variables. In addition, $T_t$ is expressed as a time trend variable, $v_i$ is a time-invariant regional fixed effect, and $u_{it}$ is an independent and identically distributed random error term. $\alpha_j (j = 0, 1, 2)$, $\lambda$, and $\gamma$ are the coefficients that need to be estimated.

### 4.1.2. Spatial Correlation Test

It is necessary to test whether there is a spatial correlation of agricultural sustainability development. The global Moran I (Moran's I) index is usually used for testing the correlation. The index calculation formula is:

$$I = \frac{1}{\sum\limits_{i=1}^{n}\sum\limits_{j=1}^{n} w_{ij}} = \frac{\sum\limits_{i=1}^{n}\sum\limits_{j=1}^{n} w_{ij}(x_i - \overline{x})(x_j - \overline{x})}{\sum\limits_{i=1}^{n}(x_i - \overline{x})^2 \Big/ n} \tag{2}$$

where $x_i$ and $x_j$ represent the observed values of regions $i$ and $j$, respectively, and $w_{ij}$ is the spatial weight matrix. If the Moran I value is greater than zero, there is a positive correlation between the regions. Otherwise, there is a negative correlation. When the Moran I value is equal to zero, there is no spatial correlation between the regions.

The above model is a spatial weight matrix used to express inter-regional connection and focuses on the spatial econometric model. In this paper, four kinds of matrices are constructed: geographical proximity matrix, geographical distance matrix, economic distance matrix, and reciprocal matrix of roads density difference. The first three represent geographical and economic links between regions. The latter is used to describe the inter-regional association brought about by rural road construction, which is specifically constructed as follows:

(1) Geographic Proximity Matrix

The basic assumption of the matrix is that only two adjacent areas have spatial interaction. The specific principles are as follows: 1 is assigned to two areas connected by the borders of 31 provinces and cities in China, and 0 refers to areas not connected in space. In addition, this paper adopts the Queen matrix. If two regions have common vertices or common edges, they can be considered as adjacent, and the form is as follows:

$$W_{ij}^{0-1} = \begin{cases} 1, & if \ i \neq j \\ 0, & if \ i = j \end{cases} \tag{3}$$

(2) Geographic Distance Matrix

If the relative size of the distance is considered, spatial adjacency can be described from a quantitative perspective, and the weights defined as:

$$W_{ij}^{dis} = \begin{cases} 1/d_{ij}^2, & if \ i \neq j \\ 0, & if \ i = j \end{cases} \tag{4}$$

$$d_{ij} = ar\cos\left[(\sin\phi_i \times \sin\phi_j) + (\cos\phi_i \times \cos\phi_j \times \cos(\Delta\tau))\right] \times R \tag{5}$$

where $\phi_i$ with $\phi_j$ are the latitude and longitude of a province, respectively, $\Delta\tau$ is the difference in longitude between the two provinces, and $R$ is earth's radius, which is equal to 3958.761 miles.

(3) Economic Distance Matrix

The first two matrices are spatial weight matrices in the geographic sense. With the rapid development of the transportation industry, however, the resistance of geographic space to economic activities is weakened, and the role of economic distance is more important. Per capita GDP is often used to reflect the economic development status of a specific area. A similarity in per capita GDP indicates that the economic development levels of the two places are similar. This article uses the difference in per capita GDP between provinces from 2002 to 2018 to demonstrate the economic distance between regions, calculated with the following equation:

$$W_{ij}^{pgdp} = \begin{cases} 1/|pgdp_i - pgdp_j|, & if \ i \neq j \\ 0, & if \ i = j \end{cases} \tag{6}$$

(4) Reciprocal Road Infrastructure Density Matrix

The construction of transportation infrastructure can reduce transportation costs and strengthen connections between regions. This study constructs a road infrastructure density spatial weight matrix to explore the spatial correlation of sustainable agricultural development in the context of transportation improvement. The equation is:

$$W_{ij}^{road} = \begin{cases} 1/|road_i - road_j|, & if \ i \neq j \\ 0, & if \ i = j \end{cases} \tag{7}$$

### 4.1.3. Spatial Econometrics Model

Because of the neglect of spatial relevance in the benchmark model, this paper plans to build a spatial econometric model to analyze this problem further. The spatial econometric model mainly includes the following three forms:

$$\text{SAR}: \ ASI_{it} = \rho \sum_{j=1}^{31} w_{ij} ASI_{it} + \alpha_1 road_{it} + Z_{it}\lambda + \gamma T_t + v_i + u_{it} \tag{8}$$

$$\text{SEM}: \ ASI_{it} = \alpha_0 + \alpha_1 road_{it} + \alpha_2 X_{it} + Z_{it}\lambda + \gamma T_t + v_i + u_{it}, \ u_{it} = \lambda w_{ij} u_{it} + \varepsilon \tag{9}$$

$$\text{SDM}: \ ASI_{it} = \rho \sum_{j=1}^{31} w_{ij} ASI_{it} + \alpha_1 road_{it} + \beta_1 \sum_{j=1}^{31} w_{ij} road_{it} + Z_{it}\lambda + \beta_2 \sum_{j=1}^{31} w_{ij} Z_{it} + \gamma T_t + v_i + u_{it} \tag{10}$$

where $\sum_{j=1}^{31} w_{ij} ASI_{it}$ is the spatial spillover effect of current regional agricultural sustainable development indicators, that is, the impact of local agricultural development on surrounding areas. The spatial econometric model can be set to the Spatial Autoregressive model (SAR), the Spatial Error Model (SEM), and the Spatial Durbin Model (SDM), depending on the source of spatial correlation. In the first form, the SAR, the model pertains to the dependent variable. In the second form, the SEM, the model is limited to the error term. The last form, the SDM, provides spatial lag values for both the dependent and independent variables. Then, we use Lagrange multiplier (LM) and Robust—LM tests of the ordinary least squares (OLS) model to verify whether we should consider the SDM model first. Moreover, the Hausman test results indicate whether the SDM model is suitable for fixed effect.

### 4.2. Variables

#### 4.2.1. Dependent Variable

(1) The Construction of Regional Agricultural Sustainable Development Index System

The dependent variable is the regional agricultural sustainable development index. At present, scholars have studied the sustainable development of agriculture from different angles. Scholars have made many explorations on sustainable agricultural development. Then, they construct the sustainable agricultural development indicators based on the theory. The whole process obeys the principles of scientificity, systematicness, sustainability, and operability. Xu [55] constructed a sustainable development index system from five systems: agricultural resources, agricultural development, environment and ecosystem, rural society, and scientific education management. Sajjad [56] constructed a sustainable livelihood security index from ecological security, economic efficiency, and social equity to analyze Bihar's temporal and spatial changes. Li [57] comprehensively considered the sustainable development of agriculture from three dimensions of society, economy, and system. Zhang [58] separated the index into five dimensions: population, society, economy, resource, and environment. According to the agricultural sustainable development goals and related literature, this paper selects five indicators of population, society, economy, resources, and environment to establish the index system of China's sustainable agricultural development to explore the sustainable development of China's agriculture (see Table 3).

**Table 3.** Evaluation index system of agricultural sustainable development in China.

| Primary-Level Indicators | Secondary-Level Indicators | Calculation Method/Yearbook Index Selection | Unit | Attribute |
|---|---|---|---|---|
| Population system sustainability | Education level of rural residents | (population with primary school education level $\times$ 6+ junior high school $\times$ 9+ senior high school $\times$ 12+ junior college and above $\times$ 16)/rural population | year | + |
| | Natural population growth rate | Natural population growth rate | % | − |
| | Regional population density | Population/area | people/10 k (km)$^2$ | − |
| Social system sustainability | Rural per capita electricity consumption | Rural electricity consumption/rural population | kwh/person | + |
| | Per capita housing area of villagers | Per capita housing area of rural residents | m$^2$/person | + |
| | Engel coefficient of rural residents | Food consumption expenditure/average consumption expenditure of rural residents | % | − |
| Economic system sustainability | Gross agricultural product per capita | The gross output value of agriculture, forestry, animal husbandry, and fishery/rural population | CNY/person | + |
| | Per capita net income of rural residents | Per capita net income of rural residents | CNY/person | + |
| | Agricultural fixed assets investment | Agricultural fixed assets investment | CNY 100 million | + |
| | Agricultural output value per unit sown area | Gross agricultural output value/sown area | CNY/hm$^2$ | + |
| Resource system sustainability | Per capita cultivated land area | Cultivated land area/population | hm$^2$/person | + |
| | Agricultural land productivity | Total grain output/cultivated land area | kg/hm$^2$ | + |
| | Total mechanical power per unit cultivated land area | Total power of agricultural machinery/cultivated land area | kW/hm$^2$ | + |
| | Agricultural water consumption | Agricultural water consumption | 1 billion m$^3$ | + |
| | Effective irrigation rate | Effective irrigated area/cultivated land area | % | + |
| Environmental system sustainability | Use intensity of chemical fertilizer | Fertilizer usage/cultivated land area | kg/hm$^2$ | − |
| | Use intensity of pesticides | Pesticide usage/cultivated land area | kg/hm$^2$ | − |
| | Use strength of mulch film | Usage of agricultural plastic film/cultivated land area | kg/hm$^2$ | − |
| | Soil and water loss control area | Soil and water loss control area | 1000 hm$^2$ | + |
| | Forest coverage | Forest coverage | % | + |
| | Agricultural disaster rate | Agricultural disaster area/cultivated land area | % | − |

(2) Calculation Method of Regional Agricultural Sustainable Development Index

Some scholars have adopted subjective weighting methods on sustainable agricultural development research methods, such as the Delphi method, analytic hierarchy process, etc. The above methods use personal judgment in the evaluation process, which determines the weight of each factor with significant subjective characteristics. In this paper, we use the entropy method, which is proposed to objectively weigh and calculate the sustainable development index of agriculture in China. This method is meant to measure the discrete degree of a specific index. The greater the dispersion degree, the greater the influence of a certain index on the comprehensive evaluation, and vice versa. The particular calculation steps are as follows:

① Standardized processing

If the index is the positive index, its calculation formula is:

$$x_{ij} = \frac{a_{ij} - \min a_{ij}}{\max a_{ij} - \min a_{ij}} \tag{11}$$

If the index is negative, its calculation formula is:

$$x_{ij} = \frac{\max a_{ij} - a_{ij}}{\max a_{ij} - \min a_{ij}} \tag{12}$$

where $a_{ij}$ is the indicator value of the item $j$ and region $i$ without standardization. $x_{ij}$ is the value of the item $j$ and region $i$ after standardization.

② Calculate the proportion of the value in the item $j$ and region $i$ to the total value of each region:

$$P_{ij} = x_{ij} \bigg/ \sum_{i=1}^{m} x_{ij} \tag{13}$$

③ Calculate the entropy value of the item $j$:

$$E_j = -\frac{\sum\limits_{i=1}^{m} P_{ij} \ln P_{ij}}{\ln m} (j = 1, 2, 3, \cdots, n) \tag{14}$$

where $m$ is the number of research units. When $P_{ij} = 0$, there exists $P_{ij} \ln P_{ij} = 0$.

④ Calculate the dynamic weight of each index:

$$W_j = (1 - E_j) \bigg/ \left( n - \sum_{j=1}^{n} E_j \right) \tag{15}$$

where $j$ is the number of indicators. The higher the weight of indicators, the greater the impact on sustainable agricultural development.

⑤ Calculate the sustainable development ability of each system. Firstly, the index weights in each system are standardized:

$$W_{kj}^* = W_j \bigg/ \sum_{j=1}^{r} W_j \tag{16}$$

Then, calculate the sustainable development ability of each system:

$$S_{ki} = \sum_{j=1}^{r} W_{kj}^* x_{ij} \tag{17}$$

where $k$ represents different systems: $k = 1, 2, 3, 4, 5$, referring to the population, social, economic, resource, and environmental systems, respectively. $S_{ki}$ represents the sustainable development ability of the system in region $i$ and system $k$. $r$ is the number of indicators contained in each system.

⑥ Calculate agricultural sustainable development index

The weight of sustainability of each system is:

$$W_k = \sum_{j=1}^{r} W_j \tag{18}$$

The following function calculates the agricultural sustainable development index:

$$ASI_i = \sum_{j=1}^{5} W_k S_{ki} \tag{19}$$

where $ASI_i$ is the regional agricultural sustainable development index in the region $i$. The range of values $ASI_i$ is $[0, 1]$. The larger the $ASI_i$ is, the stronger the comprehensive ability of sustainable development of agriculture in China.

### 4.2.2. Core Explanatory Variables

According to the introduction of China's graded highways in the second part, it can be found that third-class and fourth-class roads are channels that directly promote the connection between urban and rural areas, and thus have a direct impact on agriculture. In addition, the second-class road connects the two suburban areas, which has a specific indirect effect on agricultural development. Therefore, this paper selects the following two indicators as proxy variables for rural road construction: the first is the sum of the density of the secondary, tertiary, and fourth-level highways, which is recorded as road1. The second is the sum of the density of the third- and fourth-class roads, which is recorded as road2. The second indicator excludes secondary roads, and mainly explores whether secondary roads have indirect impacts.

### 4.2.3. Control Variables

In order to reduce the deviation of missing variables, this paper refers to the previous literature and controls the following variables:

(1) Urban–rural income gap:

According to the urban–rural income gap, there are several methods to measure the gap, such as the Theil index and Gini coefficient. Because the Gini coefficient is particularly sensitive to the change of middle income, this paper uses the Theil index of urban and rural income, and the specific formula is as follows:

$$Theil_{it} = \sum_{i=1}^{2} \left( \frac{Y_{ij,t}}{Y_{i,t}} \right) \ln \left( \frac{Y_{ij,t}}{Y_{i,t}} \Big/ \frac{Z_{ij,t}}{Z_{i,t}} \right) \tag{20}$$

where $j$ represents the region, the number 1 refers to the urban places, and the number 2 represents the rural places. $Y_{ij,t}$ is the average disposable income in provinces $i$, areas $j$, and years $t$. $Y_{i,t}$ is the average disposable income in provinces $i$ and years $t$. $Z_{ij,t}$ is the total population in provinces $i$, areas $j$, and years $t$. $Z_{i,t}$ is the total population in provinces $i$ and years $t$.

(2) Technology level: expressed by the ratio of turnover of the technology market to GDP.
(3) Industrial structure: The industrial structure gradually changes to the secondary and tertiary industries. This paper uses the tertiary industry's added value ratio and the secondary industry's value as a measure index.

(4)　Regional GDP: GDP is often used to measure a region's economic development. In order to control the regional development level, we select the logarithmic value of regional GDP instead.

(5)　Urbanization rate: expressed by the proportion of the urban resident population.

(6)　Social security level: The ratio of social security expenditure to GDP is available.

(7)　Degree of opening to the outside world: Open cities can attract a large amount of foreign capital. We select two variables: foreign direct investment and net export.

(8)　Fiscal expenditure level: measured by the ratio of fiscal expenditure to GDP.

(9)　Human capital level: Education is the primary way to improve human capital, so we select the average education level of each region to control the model. The variable is expressed by the average years of education. The calculation formula is: education level = (population with primary school education level × 6+ junior high school × 9+ senior high school × 12+ junior college and above × 16)/sampled population over six years old.

We treat the variables, such as per capita disposable income, technology market turnover, and social security expenditure at constant prices, taking 2001 as the base period to eliminate the influence of inflation. The related data above come from the EPS data analysis platform, China Statistical Yearbook, China Compendium of Statistics 1949–2008, and statistical yearbooks of all provinces. The descriptive statistics of the main variables are shown in Table 4.

**Table 4.** Descriptive statistics of the main variables.

| Variable | Symbol | Obs | Mean | Std | Min | Max |
| --- | --- | --- | --- | --- | --- | --- |
| Regional agricultural sustainable development indicators | ASI | 527 | 0.1465 | 0.0763 | 0.0449 | 0.4303 |
| Rural highway construction index | road1 | 527 | 0.6046 | 0.4331 | 0.0068 | 1.8912 |
| | road2 | 527 | 0.5234 | 0.3644 | 0.0063 | 1.4724 |
| Income gap between urban and rural areas | theil | 527 | 1.9859 | 1.0535 | 0.7359 | 6.7904 |
| Technical level | tech | 527 | 0.0101 | 0.0215 | 0.0000 | 0.1635 |
| Industrial structure | indus | 527 | 1.0170 | 0.5390 | 0.4944 | 4.3476 |
| Gross Regional Product | lnGDP | 527 | 8.8477 | 1.1342 | 5.0800 | 11.1252 |
| Urbanization rate | lnurban | 527 | 3.8732 | 0.3098 | 2.9855 | 4.4954 |
| Social security expenditure | lnsocailcare | 527 | 4.9451 | 1.2195 | 1.1811 | 6.9730 |
| Degree of opening to the outside world | inexport | 527 | 0.3129 | 0.3936 | 0.0167 | 1.8910 |
| | lnfdi | 527 | 13.8785 | 1.8661 | 7.4137 | 18.0191 |
| Expenditure | financial | 527 | 0.2374 | 0.1825 | 0.0534 | 1.3792 |
| Human capital | lneduc | 527 | 2.1299 | 0.1595 | 1.3187 | 2.5301 |

## 5. Results and Discussion

### 5.1. China's Regional Agricultural Sustainable Development

According to the above analysis, this paper calculates China's regional agricultural sustainable development index and other provinces from 2002 to 2018. At the same time, to know whether there are differences in sustainable agricultural development in different regions, this paper also calculates the coefficient of variation to explore the balance of regional agricultural sustainable development in China.

The coefficient of variation is a commonly used index to measure the relative difference of an attribute in the whole study area between years. The larger the value, the greater the relative difference of an attribute in the study area, and vice versa. Its calculation formula is as follows:

$$S_t = \sqrt{\sum_{i=1}^{n} \left( ASI_{it} - \overline{ASI_t} \right)^2 \Big/ n} \tag{21}$$

$$CV_t = S_t / \overline{ASI_t} \tag{22}$$

where $S_t$ is the standard deviation. $\overline{ASI_t}$ is China's agricultural sustainable development index in year $t$. $n$ is the number of provinces (cities, districts). $ASI_{it}$ represents the agricultural sustainable development index in region $i$ and year $t$. $CV_t$ represents the coefficient of variation.

Figure 5 shows the evolution of China's agricultural sustainable development index from 2002 to 2018. The changing trend of the agricultural sustainable development index shows a trend of first rising, then falling, and then rising again. As for the coefficient of variation of the agricultural sustainable development index, its coefficient shows a decreasing trend. It can be seen that China's regional agricultural development is moving towards sustainability and regional balance.

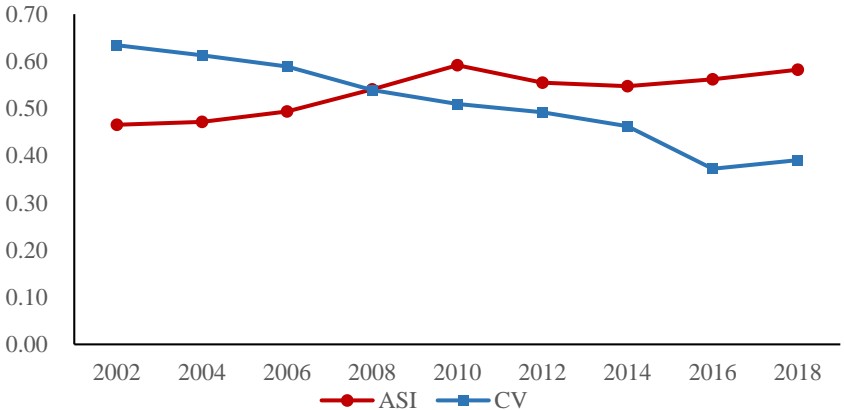

**Figure 5.** Index and coefficient of variation of regional agricultural sustainable development in China (2002–2018).

### 5.2. Exploratory Spatial Data Analysis

In this paper, the Moran index is calculated using four spatial weight matrices: geographical proximity matrix, geographical distance matrix, economic distance matrix, and road density difference matrix. It can be seen from the following results that the Moran index is significantly positive under each spatial weight matrix before 2016, which indicates that the regional agricultural sustainable development index of each region has spatial correlation (detailed results are shown in Table 5). Since 2016, the Chinese government has promoted the development of characteristic regional agriculture, allowing regions to focus on developing their characteristics and reducing inter-regional relevance. In addition, the Moran index under the highway density matrix is larger than the other three, which indicates that the highway construction makes the connection between regions closer. From the results, it can be seen that exploring the relationship between rural highway construction and regional agricultural sustainable development will lead to biased estimation results.

### 5.3. Main Results
#### 5.3.1. Rural Road Construction and Sustainable Development of Regional Agriculture

The regression results of FE, SDM, SAR, and SEM are listed in Table 6. It can be seen from the LM test that it is more appropriate to choose the Spatial Durbin Model; that is, to add a spatial lag term and spatial error term into the model simultaneously. On the whole, rural road construction can significantly improve the sustainable development capacity of regional agriculture and has passed the significance level of 1%. Over the years, China's unique urban–rural dual structure has brought many restrictions to the sustainable development of China's economy. It also has a significant impact on the sustainable development of agriculture. On the one hand, rural road construction promotes rural surplus labor transfer to cities and improves rural labor productivity. On the other hand, it can also enable many resources to flow from cities to rural areas, such as agricultural machinery and equipment, high-quality seeds, chemical fertilizers, etc. These will contribute to the sustainable development of agriculture.

**Table 5.** Moran's I value of Theil index of agricultural sustainable development.

| Year | Geographic Proximity Matrix | Geographic Distance Matrix | Economic Distance Matrix | Road Density Matrix |
|---|---|---|---|---|
| 2002 | 0.3090 *** (3.2710) | 0.3300 *** (4.6000) | 0.3770 *** (2.7470) | 0.6200 *** (4.2110) |
| 2003 | 0.3020 *** (3.2370) | 0.3370 *** (4.7290) | 0.3810 *** (2.7970) | 0.6330 *** (4.3350) |
| 2004 | 0.2890 *** (3.1240) | 0.3280 *** (4.6250) | 0.3840 *** (2.8280) | 0.5930 *** (4.1330) |
| 2005 | 0.3100 *** (3.3520) | 0.3360 *** (4.7820) | 0.4140 *** (3.0610) | 0.6380 *** (4.4850) |
| 2006 | 0.2830 *** (3.1030) | 0.3280 *** (4.6990) | 0.4120 *** (3.0610) | 0.7800 *** (5.7840) |
| 2007 | 0.2620 *** (2.9240) | 0.3140 *** (4.5580) | 0.4110 *** (3.0830) | 0.8230 *** (6.1910) |
| 2008 | 0.2390 *** (2.7110) | 0.3070 *** (4.4870) | 0.4130 *** (3.1200) | 0.8460 *** (6.2970) |
| 2009 | 0.1790 ** (2.0550) | 0.2730 *** (3.9380) | 0.4180 *** (3.0690) | 0.7590 *** (5.0820) |
| 2010 | 0.1780 ** (2.0950) | 0.2840 *** (4.1660) | 0.4270 *** (3.1980) | 0.7820 *** (5.4220) |
| 2011 | 0.1240 * (1.5540) | 0.2480 *** (3.6820) | 0.3830 *** (2.8800) | 0.7460 *** (5.2360) |
| 2012 | 0.1120 * (1.4260) | 0.2540 *** (3.7290) | 0.4120 *** (3.0610) | 0.7890 *** (5.5420) |
| 2013 | 0.1030 * (1.3070) | 0.2410 *** (3.4720) | 0.3950 *** (2.8670) | 0.7280 *** (5.0970) |
| 2014 | 0.1110 * (1.3740) | 0.2420 *** (3.4740) | 0.4100 *** (2.9600) | 0.6980 *** (4.9720) |
| 2015 | 0.1240 * (1.5140) | 0.2490 *** (3.5790) | 0.4320 *** (3.1260) | 0.6640 *** (4.8230) |
| 2016 | −0.0450 (−0.1210) | 0.0110 (−0.6010) | 0.0620 (−0.6820) | −0.0060 (−0.1980) |
| 2017 | −0.0510 (−0.1770) | 0.0020 (0.4600) | 0.0420 (0.3010) | −0.019 (−0.1000) |
| 2018 | −0.0500 (−0.1780) | −0.0180 (−0.2190) | 0.0070 (0.2970) | −0.0700 (−0.2710) |

Notes: The *t*-values are in parentheses. *** means the significance level is 1%; ** means the significance level is 5%; and * means the significance level is 10%.

From the control variables: (1) The income gap between urban and rural residents will significantly reduce the sustainable development level of agriculture. On the one hand, the widening income gap between urban and rural areas has prompted many rural laborers to move to cities. The agricultural economy has not developed, thus inhibiting the sustainable development of agriculture. On the other hand, the large income gap between urban and rural areas will encourage rural residents to overuse chemical fertilizers to increase their output. Environmental sustainability has also been compromised. (2) The technological development level and industrial structure of each province negatively impact the sustainable development of agriculture. Since most of the technical inputs of provinces are put into the secondary industry, this may have a specific crowding-out effect on agricultural development to a certain extent. It can be seen from the above results that there is continuous development of the urban economy. The widening gap between urban and rural areas has inhibited the sustainable development of agriculture.

**Table 6.** Basic econometric estimation results.

| VAR | (1) | | (2) | | (3) | | (4) | |
|---|---|---|---|---|---|---|---|---|
| | OLS | OLS | SDM | SDM | SAR | SAR | SEM | SEM |
| road1 | 0.0435 *** | | 0.0574 *** | | 0.0404 *** | | 0.0434 *** | |
| | (3.9134) | | (4.6459) | | (3.5586) | | (3.7240) | |
| road2 | | 0.0300 ** | | 0.0523 *** | | 0.0389 *** | | 0.0419 *** |
| | | (2.1845) | | (4.0094) | | (3.2193) | | (3.3954) |
| theil | −0.0270 *** | −0.0325 *** | −0.0070 | −0.0068 | −0.0217 *** | −0.0209 ** | −0.0244 *** | −0.0237 *** |
| | (−3.3344) | (−2.8425) | (−0.8225) | (−0.7960) | (−2.6238) | (−2.5175) | (−2.9086) | (−2.8301) |
| tech | −1.0364 *** | −0.7620 *** | −1.0096 *** | −1.0062 *** | −0.8597 *** | −0.8613 *** | −0.8638 *** | −0.8764 *** |
| | (−5.4999) | (−3.7056) | (−4.7556) | (−4.7109) | (−4.4634) | (−4.4557) | (−4.4504) | (−4.4998) |
| indus | −0.0575 *** | 0.1160 *** | −0.0212 | −0.0198 | −0.0552 ** | −0.0547 ** | −0.0579 *** | −0.0573 *** |
| | (−2.7214) | (2.6191) | (−0.9891) | (−0.9168) | (−2.5618) | (−2.5287) | (−2.6750) | (−2.6380) |
| lnGDP | −0.0285 ** | 0.0402 *** | 0.0094 | 0.0085 | −0.0284 * | −0.0285 * | −0.0330 ** | −0.0328 ** |
| | (−1.9884) | (4.0912) | (0.5947) | (0.5328) | (−1.9397) | (−1.9374) | (−2.2168) | (−2.1956) |
| lnurban | −0.1348 *** | −0.1627 *** | −0.0413 | −0.0440 | −0.1062 *** | −0.1045 *** | −0.1169 | −0.1166 *** |
| | (−3.6836) | (−3.4062) | (−1.1012) | (−1.1584) | (−2.8323) | (−2.7741) | (−3.0854) | (−3.0644) |
| lnsocialcare | −0.0001 | −0.0106 | −0.0024 | −0.0031 | −0.0007 | 0.00002 | 0.0003 | 0.0008 |
| | (−0.0117) | (−1.2457) | (−0.3839) | (−0.4925) | (−0.1204) | (−0.0037) | (0.0516) | (0.1279) |
| inexport | −0.0048 | −0.0283 * | −0.0247 ** | −0.0259 | −0.0072 | −0.0082 | −0.0060 | −0.0073 |
| | (−0.3947) | (−1.7082) | (−1.9770) | (−2.0522) | (−0.5743) | (−0.6473) | (−0.4781) | (−0.5726) |
| lnfdi | −0.0066 *** | −0.0109 *** | −0.0058 ** | −0.0060 ** | −0.0057 ** | −0.0057 ** | −0.0058 ** | −0.0058 ** |
| | (−2.7231) | (−2.9159) | (−2.3629) | (−2.4213) | (−2.3121) | (−2.2823) | (−2.3049) | (−2.2989) |
| financial | −0.0339 | 0.1154 ** | 0.0235 | 0.0199 | −0.0183 | −0.0225 | −0.0221 * | −0.0266 |
| | (−0.9691) | (2.5574) | (0.6567) | (0.5561) | (−0.5125) | (−0.6298) | (−0.6148) | (−0.7390) |
| lneduc | −0.0829 * | 0.3061 *** | −0.0568 | −0.0604 | −0.0829 * | −0.0838 * | −0.0828 | −0.0843 * |
| | (−1.7588) | (6.0922) | (−1.2599) | (−1.3335) | (−1.7252) | (−1.7373) | (−1.7165) | (−1.7403) |
| W*X | N | N | Y | Y | N | N | N | N |
| rho | | | 0.0409 | 0.0278 | 0.3157 *** | 0.3118 *** | | |
| | | | (0.5559) | (0.3755) | (4.8733) | (4.7974) | | |
| R-sq | 0.1366 | 0.3876 | 0.8967 | 0.8955 | 0.8782 | 0.8776 | 0.8742 | 0.8738 |
| LM-lag | 9.2790 *** | 8.9953 *** | | | | | | |
| R-LM-Lag | 40.9426 | 42.9929 *** | | | | | | |
| LM-error | 1.6706 *** | 1.4891 | | | | | | |
| R-LM-error | 33.3342 *** | 35.4867 *** | | | | | | |
| Hausman Test | 181.0322 *** | 74.1387 *** | | | | | | |
| Obs | 527 | 527 | 527 | 527 | 527 | 527 | 527 | 527 |

Notes: The *t*-values are in parentheses. *** means the significance level is 1%; ** means the significance level is 5%; and * means the significance level is 10%. In these models, we chose the geographic distance matrix as the spatial weighted matrix.

Lesage and Pace [59] put forward the concepts and decomposition methods of direct effect, indirect effect, and total effect to solve uncertain coefficients in the spatial econometric model. The direct effect indicates the influence of the independent variable on the dependent variable in a particular area, including model coefficient and feedback effect. Feedback effect refers to the impact of independent variables in one region on the dependent variable in the other regions. Meanwhile, other regions influence the interpreted variables in this region in turn. From this point of view, the decomposition of the spatial spillover effect is more accurate and more realistic than looking at model coefficients directly. The indirect effect refers to the influence of local explanatory variables on interpreted variables in other regions.

In contrast, total effect refers to the average impact of local explanatory variables on all areas. As shown from Table 7, the direct effect of rural road construction is all positive, meaning rural road construction significantly promotes the sustainable development of local agriculture. Moreover, the impact of secondary and tertiary highways is far more extensive than that of the fourth-class roads. Because secondary roads and tertiary roads promote the connection between cities and county towns, the communication scope between regions is broader. Thus, there will be more opportunities for development in rural

areas, leading to a more significant contribution to agricultural development. In addition, the construction of rural roads also has a negative indirect effect. The continuous improvement of local rural roads will negatively impact the sustainable development of agriculture in the surrounding areas. Rural road construction is a systematic project. In the process of rural road construction, we should consider the local impact of roads construction, and measure its external effects to realize the sustainable development of the whole region.

**Table 7.** Direct, indirect, and total effects of the rural road construction.

| Effects | road1 | road2 | Second | Third | Fourth |
|---|---|---|---|---|---|
| Direct Effects | 0.0585 *** (4.8886) | 0.0528 *** (4.1072) | 0.2271 *** (3.3186) | 0.3553 *** (2.8991) | 0.0509 *** (3.9556) |
| Indirect Effects | −0.0653 * (−1.9544) | −0.0466 (−1.3192) | −0.3156 * (−1.8710) | 0.2706 (0.9016) | −0.0460 (−1.2381) |
| Total Effects | −0.0068 (−0.2162) | 0.0062 (0.1873) | −0.0885 (−0.4557) | 0.6259 * (1.8605) | 0.0049 (0.1374) |

Notes: The *t*-values are in parentheses. *** means the significance level is 1%; ** means the significance level is 5%; and * means the significance level is 10%.

5.3.2. Rural Road Construction and Sustainable Development Ability of Population, Society, Economy, Resources, and Environment

This paper also uses rural road construction as the core explanatory variable and the five dimensions of sustainability index as the explained variable for regression to explore the impact of rural road construction on the sustainable development capacity of different dimensions. It can be seen from Table 8 that the construction of rural roads has significantly promoted the sustainable development of the three dimensions of population, society, and resources. From the results, we could find that the construction of rural roads has promoted the flow of population, capital, and commodities, thereby increasing rural human capital and rural labor productivity. At the same time, rural roads positively impact the sustainable development of the economic dimension, but the impact is not significant. In addition, however, the construction of rural roads significantly reduces the environmental dimension of sustainable development. Although the construction of rural roads has facilitated the inflow of rural residents to cities, obtained more resources, and promoted its population and social development, it has also caused specific damage to the local environment, such as excessive use of fertilizers. The reason is that, on the one hand, lower transportation costs make the logistics costs of chemical fertilizers and pesticides transported to rural areas lower, and their prices also drop so that rural residents can buy more chemical fertilizers.

On the other hand, road construction brings urban and rural residents to communicate more often. However, the massive gap between urban and rural areas has prompted rural residents to increase fertilizers for more production. In addition, during the construction and operation of rural roads, certain negative externalities will also be generated on the surrounding farmland, which will affect the quality of the land. Based on this, the construction of rural roads has a greater impact on environmental sustainability. Therefore, the government needs to pay attention to the negative effects of rural road construction on the environment and formulate relevant policies to guide farmers to use chemical fertilizers rationally to achieve sustainable agricultural development.

**Table 8.** Regression Results of Rural Road Construction and Sustainable Development Ability of Population, Society, Economy, Resources, and Environment.

| VAR | (1) Population | | (2) Social | | (3) Economic | | (4) Resource | | (5) Environment | |
|---|---|---|---|---|---|---|---|---|---|---|
| road1 | 0.0261 *** (2.8263) | | 0.0147 *** (4.5197) | | 0.0017 (0.4847) | | 0.0078 *** (4.2352) | | −0.0040 *** (−4.3693) | |
| road2 | | 0.0293 *** (3.0263) | | 0.0075 ** (2.1465) | | 0.0029 (0.8104) | | 0.0084 *** (4.4024) | | −0.0045 *** (−4.6430) |
| theil | −0.0007 (−0.1047) | −0.0013 (−0.2027) | −0.0007 (−0.3167) | 0.0002 (0.0762) | 0.0050 ** (2.1348) | 0.0048 ** (2.0512) | −0.0027 ** (−2.1357) | −0.0028 ** (−2.2511) | −0.0039 *** (−6.2724) | −0.0038 *** (−6.0755) |
| indus | −0.0228 (−1.4214) | −0.0211 (−1.3144) | 0.0060 (1.0583) | 0.0058 (1.0002) | 0.0022 (0.3711) | 0.0022 (0.3749) | −0.0062 * (−1.9593) | −0.0061 * (−1.9383) | 0.0029 * (1.7999) | 0.0028 * (1.7888) |
| lnGDP | 0.0253 ** (2.1434) | 0.0249 ** (2.1171) | −0.0042 (−1.0094) | −0.0043 (−1.0274) | 0.0016 (0.3624) | 0.0015 (0.3472) | −0.0011 (−0.4806) | −0.0014 (−0.5967) | 0.0011 (0.8965) | 0.0012 (1.0030) |
| lnurban | 0.0051 (0.1823) | 0.0006 (0.0225) | −0.0204 ** (−2.0676) | −0.0175 * (−1.7339) | 0.0216 ** (2.0780) | 0.0207 ** (1.9879) | −0.0125 ** (−2.2590) | −0.0134 ** (−2.4185) | −0.0156 *** (−5.6286) | −0.0150 *** (−5.3792) |
| lnsocialcare | 0.0037 (0.7843) | 0.0036 (0.7846) | −0.0017 (−1.0075) | −0.0019 (−1.1563) | −0.0035 ** (−2.0588) | −0.0036 ** (−2.0953) | −0.0019 ** (−2.0963) | −0.0020 ** (−2.1829) | 0.0013 *** (2.7721) | 0.0014 *** (2.9830) |
| inexport | −0.0050 (−0.5303) | −0.0064 (−0.6828) | −0.0199 *** (−6.0304) | −0.0188 *** (−5.5986) | 0.0045 (1.2983) | 0.0041 (1.1729) | −0.0002 (−0.1162) | −0.0007 (−0.3852) | 0.0016* (1.7635) | 0.0020 ** (2.1601) |
| lnfdi | −0.0024 (−1.3300) | −0.0026 (−1.4241) | −0.0027 *** (−4.1803) | −0.0027 *** (−4.1046) | −0.0006 (−0.8562) | −0.0006 (−0.8672) | −0.0004 (−1.0982) | −0.0004 (−1.1593) | 0.00004 (0.2314) | 0.00004 (0.2414) |
| financial | 0.0187 (0.6986) | 0.0188 (0.7097) | −0.0075 (−0.7918) | −0.0100 (−1.0469) | 0.0149 (1.5086) | 0.0151 (1.5409) | 0.0159 *** (3.0064) | 0.0156 *** (2.9767) | −0.0212 *** (−8.0230) | −0.0211 *** (−8.0494) |
| lneduc | −0.0287 (−0.8521) | −0.0309 (−0.9188) | −0.0019 (−0.1579) | −0.0018 (−0.1460) | −0.0037 (−0.2983) | −0.0041 (−0.3278) | −0.0010 (−0.1513) | −0.0016 (−0.2437) | 0.0037 (1.1223) | 0.0042 (1.2694) |
| W*X | Y | Y | Y | Y | Y | Y | Y | Y | Y | Y |
| rho | −0.0684 (−0.8929) | −0.0745 (−0.9718) | 0.0550 (0.7647) | 0.0265 (0.3640) | 0.0728 (0.9859) | 0.0686 (0.9282) | 0.2053 *** (2.8941) | 0.2125 *** (3.0062) | 0.3449 *** (5.5045) | 0.3631 *** (5.8737) |
| R-sq | 0.9137 | 0.9140 | 0.8631 | 0.8584 | 0.9233 | 0.9233 | 0.9543 | 0.9544 | 0.9783 | 0.9783 |
| LM-lag | 4.21 ** | 131.43 *** | 49.43 *** | 53.29 *** | 39.87 *** | 39.70 *** | 0.1663 | 0.3360 | 31.55 *** | 31.10 *** |
| R-LM-Lag | 37.79 *** | 39.78 *** | 8.52 *** | 10.89 *** | 22.99 *** | 23.91 *** | 10.44 *** | 8.82 *** | 66.64 *** | 65.79 *** |
| LM-error | 0.03 | 109.39 *** | 42.21 *** | 42.80 *** | 22.24 *** | 21.80 *** | 3.72 * | 4.01 ** | 7.62 *** | 8.05 *** |
| R-LM-error | 33.61 *** | 17.74 *** | 1.30 | 0.40 | 5.36 ** | 6.01 ** | 14.00 *** | 12.52 *** | 42.71 *** | 41.73 *** |
| Hausman Test | 50.14 *** | 51.42 *** | 56.10 *** | 65.46 *** | 34.67 * | 12.96 | 19.34 * | 21.46 ** | 26.54 | 27.06 |
| Obs | 527 | 527 | 527 | 527 | 527 | 527 | 527 | 527 | 527 | 527 |

Notes: The *t*-values are in parentheses. *** means the significance level is 1%; ** means the significance level is 5%; and * means the significance level is 10%.

### 5.3.3. Robustness Test
#### (1) Replace Spatial Weight Matrix

To ensure the robustness of the results, we also take the geographic proximity matrix, economic distance matrix, and the difference matrix of highway density as the spatial weight matrix to explore the impact of rural road construction on regional agricultural sustainable development. It can be seen from Table 9 that the coefficients of road1 and road2 are significantly positive, which indicates that rural road construction has indeed promoted the sustainable development of regional agriculture. The above results are robust.

**Table 9.** Robustness test 1: replacement space weight matrix.

| VAR | (1) | | (2) | | (3) | |
|---|---|---|---|---|---|---|
| | 0–1 | | Economic | | Highway | |
| road1 | 0.0733 *** | | 0.0542 *** | | 0.0648 *** | |
| | (4.7335) | | (4.7550) | | (4.8654) | |
| road2 | | 0.0674 *** | | 0.0511 *** | | 0.0572 *** |
| | | (4.0370) | | (4.2416) | | (3.9320) |
| theil | −0.0101 | −0.0107 | −0.0100 | −0.0099 | −0.0142 * | −0.0134 |
| | (−1.1358) | (−1.1867) | (−1.2090) | (−1.1979) | (−1.8146) | (−1.6199) |
| tech | −0.9125 *** | −0.9112 *** | −0.8765 *** | −0.8864 *** | −0.9664 *** | −0.9431 *** |
| | (−4.8120) | (−4.7679) | (−4.3058) | (−4.3141) | (−4.8483) | (−4.4912) |
| indus | −0.0413 ** | −0.0390 * | −0.0467 ** | −0.0455 ** | −0.0336 | −0.0319 |
| | (−1.9845) | (−1.8609) | (−2.2722) | (−2.1948) | (−1.6312) | (−1.4685) |
| lnGDP | −0.0277 * | −0.0276 * | 0.0069 | 0.0062 | −0.0113 | −0.0111 |
| | (−1.9200) | (−1.8996) | (0.4541) | (0.4054) | (−0.7607) | (−0.7090) |
| lnurban | −0.0664 * | −0.0713 * | −0.0547 | −0.0561 | −0.0933 *** | −0.0916 ** |
| | (−1.6686) | (−1.7744) | (−1.5135) | (−1.5365) | (−2.6661) | (−2.4758) |
| lnsocialcare | −0.0037 | −0.0032 | −0.0049 | −0.0052 | −0.0034 | −0.0036 |
| | (−0.6437) | (−0.5488) | (−0.7928) | (−0.8300) | (−0.5680) | (−0.5698) |
| inexport | −0.0025 | −0.0048 | −0.0100 | −0.0113 | −0.0238 * | −0.0237* |
| | (−0.2017) | (−0.3892) | (−0.7554) | (−0.8533) | (−1.9499) | (−1.8246) |
| lnfdi | −0.0042 * | −0.0045 * | −0.0052 ** | −0.0052 ** | −0.0052 ** | −0.0054 ** |
| | (−1.7119) | (−1.8217) | (−2.1986) | (−2.1971) | (−2.2075) | (−2.1936) |
| financial | 0.0389 | 0.0330 | 0.0112 | 0.0107 | 0.0032 | 0.0048 |
| | (1.0982) | (0.9297) | (0.3149) | (0.2998) | (0.0852) | (0.1226) |
| lneduc | −0.0507 | −0.0529 | −0.0594 | −0.0622 | −0.0401 | −0.0416 |
| | (−1.1371) | (−1.1772) | (−1.3476) | (−1.4028) | (−0.9139) | (−0.9007) |
| W*X | Y | Y | Y | Y | Y | Y |
| rho | 0.0290 | 0.0330 | 0.0130 | 0.0100 | 0.0290 | 0.0582 |
| | (0.4731) | (0.5388) | (0.2978) | (0.2281) | (0.6619) | (1.3314) |
| R-sq | 0.8893 | 0.8882 | 0.8930 | 0.8917 | 0.8951 | 0.8940 |
| LM-lag | 1.7306 | 1.6607 | 4.3245 ** | 4.0887 ** | 6.5663 ** | 6.4041 * |
| R-LM-Lag | 7.2065 *** | 7.8920 *** | 29.6096 *** | 31.2979 *** | 26.9377 *** | 28.6954 *** |
| LM-error | 0.4801 | 0.4168 | 0.5107 | 0.3820 | 1.6029 | 1.4245 |
| R-LM-error | 5.9561 ** | 6.6482 * | 25.7957 *** | 27.5912 *** | 21.l9743 *** | 23.7158 *** |
| Hausman Test | 70.1186 *** | 60.3785 *** | 31.1758 *** | 28.8741 *** | 28.3268 *** | 29.9245 *** |
| Obs | 527 | 527 | 527 | 527 | 527 | 527 |

Notes: The *t*-values are in parentheses. *** means the significance level is 1%; ** means the significance level is 5%; and * means the significance level is 10%.

#### (2) Endogenous Problems

Although the Spatial Durbin Model can solve the endogenous problems caused by the lag and missing variables of the explained variables, the method still cannot solve the endogenous problems caused by the reciprocal causality between the explanatory variables and the explained variables. Generally speaking, when these places have good economic conditions, the rural road construction will be more completed. The level of agricultural development is also relatively good. Considering the endogenous problem between rural road construction and regional agricultural development, we select the

instrumental variable method to solve it with reference to the research of Jiang [60]. In this paper, the instrumental variables of the interaction term in the nearest geographical distance between the provincial capital cities and the major cities (Beijing, Shanghai, Guangzhou, Shenzhen) and the average temperature in the same year are used to estimate. To avoid missing values, we set the distance between Beijing, Shanghai, and Guangzhou as 1 km (not 0 km). This instrumental variable is selected for the following reasons.

Firstly, the geographical distance between the provincial capital and central cities is not affected by exogenous variables, like time and economic development. At the same time, the provincial capital cities have strong radiation power over other cities in the province. When the provincial capital cities are very close to these central cities, the stronger the demand to integrate into the above-mentioned central cities, and the greater the external incentive for urban and rural infrastructure construction. Therefore, the enthusiasm for rural road construction in various regions is higher.

Secondly, China is mainly located in the subtropical and temperate climate zone. The annual temperature is relatively mild, and there are few extreme kinds of weather, such as extremely hot and cold, which will be conducive to infrastructure construction and agricultural production. Therefore, the higher the average temperature in each province, the more suitable for rural roads.

Thirdly, the interaction between geographical distance and average temperature indicates that the closer the geographical distance to the above-mentioned central cities and the higher the average temperature, the more conducive it is to rural road construction. Therefore, this variable is more suitable as an instrumental variable for rural road construction. This paper uses the Baidu map to calculate the shortest geographical distance between the provincial capital cities and the four central cities, and takes the shortest straight line distance between the two places. As for the annual average temperature of each province, the annual average temperature of each provincial capital city is replaced by the average yearly temperature of each provincial capital city.

As shown from Table 10, the Sargan value and adjoint probability of each regression over-identification test indicate no over-identification problem, and the instrumental variables are strictly exogenous. Meanwhile, the coefficients of rural roads in the following regression are significantly positive, showing that the results are robust when considering the endogenous problems.

**Table 10.** Robustness test 2: instrumental variable method.

| VAR | (1) | | (2) | |
|---|---|---|---|---|
| | FE+IV | FE+IV | SDM+IV | SDM+IV |
| road1 | 0.0596 *** | | 0.0544 *** | |
| | (4.0700) | | (3.8100) | |
| road2 | | 0.0637 *** | | 0.0598 *** |
| | | (4.0500) | | (3.8400) |
| theil | −0.0228 ** | −0.0226 ** | −0.02677 *** | −0.0271 *** |
| | (−2.4800) | (−2.4500) | (−2.7600) | (−2.8100) |
| tech | −0.9347 *** | −0.9269 *** | −1.0308 *** | −1.0259 *** |
| | (−4.5400) | (−4.4800) | (−5.2300) | (−5.1800) |
| indus | −0.0374 * | −0.0361 * | −0.0017 | −0.0107 |
| | (−1.8400) | (−1.7700) | (−0.6600) | (−0.5700) |
| lnGDP | 0.0254 ** | 0.0261 ** | 0.0218 ** | 0.0222 ** |
| | (2.2000) | (2.2500) | (2.1200) | (2.1600) |
| lnurban | −0.1117 *** | −0.1132 *** | −0.1283 *** | −0.1316 *** |
| | (−2.6200) | (−2.6400) | (−3.1400) | (−3.2200) |
| lnsocialcare | 0.0031 | 0.0031 | 0.0008 | 0.0009 |
| | (0.6600) | (0.6600) | (0.2200) | (0.2200) |
| inexport | −0.0221 * | −0.0263 ** | −0.0186 | −0.0232 * |
| | (−1.7300) | (−2.0100) | (−1.5400) | (−1.8900) |

**Table 10.** *Cont.*

| VAR | (1) | | (2) | |
|---|---|---|---|---|
| | FE+IV | FE+IV | SDM+IV | SDM+IV |
| lnfdi | −0.0079 *** | −0.0079 *** | −0.0050 ** | −0.0052 ** |
| | (−2.9600) | (−2.9500) | (−1.9600) | (−2.0300) |
| financial | 0.0259 | 0.0257 | 0.0325 | 0.0408 |
| | (0.7000) | (0.7000) | (1.2100) | (1.1500) |
| lneduc | −0.0481 | −0.0512 | 0.0033 | −0.0009 |
| | (−1.1100) | (−1.1700) | (0.0900) | (−0.0200) |
| W*X | N | N | Y | Y |
| rho | | | 0.7608 *** | 0.7658 *** |
| | | | (11.5700) | (11.6100) |
| R-sq | 0.2608 | 0.2558 | 0.4740 | 0.4677 |
| Sargan | 2.4330 | 2.4680 | 0.6820 | 0.7500 |
| | (0.2963) | (0.2910) | (0.7109) | (0.6872) |
| Hausman Test | 4.9700 | 7.4400 | 110.4800 *** | 92.2400 *** |
| Obs | 527 | 527 | 527 | 527 |

Notes: The *t*-values are in parentheses. *** means the significance level is 1%; ** means the significance level is 5%; and * means the significance level is 10%.

## 6. Further Analysis

### 6.1. The Lagging Effect of Rural Road Construction on Sustainable Agricultural Development

Since 2002, China has made great efforts to develop rural road construction. In 2003, the Party group of the Ministry of Transport put forward the idea of building roads to serve the rural areas and serve the urbanization, thus letting the farmers walk on the asphalt road and cement road. Then, China started the largest rural road construction project since founding the people's Republic of China. In 2006, the Party committee of the Ministry of Transport proposed to do an excellent job in three services. It serves the national economy and social development, constructing a new socialist countryside and a safe and convenient trip. With the approval of the State Council, the "Eleventh Five-Year Plan" and "5 Billion Project" of rural road construction were officially launched in 2006. In 2008, to promote China's economic development, the state launched the Four Trillion Investment Plan, in which most of the funds were invested in the construction of transportation infrastructure. It is necessary to explore whether the investment projects in 2008 promoted the sustainable development of agriculture more quickly because of the vigorous development of the country's transportation infrastructure after 2008. Therefore, this article uses 2008 as the time node to explore the impact of rural road construction on the sustainable development of agriculture in the two periods. It can be seen from Table 11 that from 2002–2007, the effect of rural road construction was significantly lower than that of 2008–2018. After excluding the impact of secondary roads, the construction of rural roads in 2002–2007 did not significantly impact the sustainable development of local agriculture. We could also find that the impact of rural road construction has a certain lag from the results. Meanwhile, with the continuous improvement of rural road construction, the greater its impact on the sustainable development of agriculture. In addition, this paper also made a regression of one period and two periods of lag to explore whether the effects of rural road construction are lagging. From the regression results, it can be seen that the lag period and the two lag periods are significant and more remarkable than the effect of the current period. The lag of two periods is lower than the effect of the one period of lag. Thus, the effect of rural road construction has a lag in time, and the effect has the characteristic of decreasing marginally.

**Table 11.** Regression results of heterogeneity analysis based on different periods.

| VAR | (1) 2002–2007 | | (2) 2008–2018 | | (3) Lag (1) | | (4) Lag (2) | |
|---|---|---|---|---|---|---|---|---|
| road1 | 0.0131 * (1.7540) | | 0.0921 * (2.7316) | | 0.0593 *** (4.4660) | | 0.0520 *** (3.5901) | |
| road2 | | 0.0108 (1.3965) | | 0.0765 ** (2.2477) | | 0.0537 *** (3.8171) | | 0.0465 *** (3.0414) |
| theil | 0.0116 ** (2.3872) | 0.0120 ** (2.4782) | 0.0035 (0.1839) | 0.0025 (0.1300) | −0.0087 (−0.9215) | −0.0088 (−0.9280) | −0.0087 (−0.8225) | −0.0095 (−0.8935) |
| tech | −0.8579 *** (−3.1093) | −0.8376 *** (−3.0278) | −0.6272 ** (−1.8179) | −0.6525 * (−1.8832) | −0.9719 *** (−4.3575) | −0.9759 *** (−4.3460) | −0.9293 *** (−3.9039) | −0.9392 *** (−3.9125) |
| indus | 0.0133 * (1.7052) | 0.0130* (1.6640) | −0.2281 *** (−3.2764) | −0.2243 *** (−3.2212) | −0.0242 (−1.0815) | −0.0220 (−0.9759) | −0.0282 (−1.1833) | −0.0264 (−1.0990) |
| lnGDP | −0.0154 (−1.3671) | −0.0153 (−1.3551) | −0.0176 (−0.5042) | −0.0117 (−0.3346) | 0.0187 (1.0738) | 0.0173 (0.9874) | 0.0273 (1.3951) | 0.0254 (1.2963) |
| lnurban | 0.0511 ** (2.4302) | 0.0529 ** (2.5257) | −0.0959 (−0.8718) | −0.1129 (−1.0224) | −0.0563 (−1.2765) | −0.0610 (−1.3707) | −0.0671 (−1.2298) | −0.0763 (−1.3884) |
| lnsocialcare | −0.0043 (−1.5106) | −0.0041 (−1.4468) | −0.0065 (−0.3800) | −0.0073 (−0.4252) | −0.0024 (−0.3428) | −0.0035 (−0.4974) | −0.0021 (−0.2653) | −0.0034 (−0.4308) |
| inexport | −0.0021 (−0.1894) | 0.0002 (0.0192) | −0.0092 (−0.3651) | −0.0075 (−0.2954) | −0.0195 (−1.4439) | −0.0218 (−1.5953) | −0.0147 (−0.9768) | −0.0166 (−1.0892) |
| lnfdi | −0.0010 (−0.5442) | −0.0011 (−0.5749) | −0.0064 (−1.5310) | −0.0064 (−1.5175) | −0.0063 ** (−2.4148) | −0.0064 ** (−2.4374) | −0.0064 ** (−2.2282) | −0.0064 ** (−2.2100) |
| financial | −0.0120 (−0.3506) | −0.0134 (−0.3931) | 0.0468 (0.5629) | 0.0386 (0.4661) | 0.0228 (0.5873) | 0.0177 (0.4550) | 0.0193 (0.4441) | 0.0128 (0.2943) |
| lneduc | 0.0010 (0.0404) | 0.0007 (0.0293) | −0.0731 (−1.0859) | −0.0718 (−1.0593) | −0.0509 (−1.0557) | −0.0542 (−1.1188) | −0.0595 (−1.1227) | −0.0615 (−1.1560) |
| W*X | Y | Y | Y | Y | Y | Y | Y | Y |
| rho | 0.5970 *** (7.3896) | 0.6037 *** (7.5557) | 0.0416 (0.4451) | 0.0241 (0.2562) | 0.0394 (0.5186) | 0.0284 (0.3718) | 0.0391 (0.4965) | 0.0292 (0.3683) |
| R-sq | 0.9939 | 0.9939 | 0.8755 | 0.8741 | 0.8935 | 0.8923 | 0.8878 | 0.8868 |
| LM-lag | 36.0863 *** | 35.6303 *** | 0.3903 | 50.7923 *** | 7.5499 *** | 7.3384 *** | 5.4177 ** | 5.2513 ** |
| R-LM-Lag | 42.3104 *** | 38.5913 *** | 4.9451 ** | 30.4347 *** | 33.5790 *** | 35.4401 *** | 27.9912 *** | 29.0058 *** |
| LM-error | 15.0166 *** | 15.2140 *** | 0.0199 | 11.5403 *** | 1.3805 | 1.2352 | 0.8714 | 0.7793 |
| R-LM-error | 21.2588 *** | 18.1749 *** | 4.5746 ** | 22.9035 *** | 27.4096 *** | 29.3369 *** | 23.4450 *** | 24.5339 *** |
| Hausman Test | 5.3689 | 5.0786 | 51.4315 *** | 4.0091 ** | 43.4184 *** | 39.7473 ** | 31.5860 | 28.3307 |
| Obs | 186 | 186 | 341 | 341 | 496 | 496 | 465 | 465 |

Notes: The *t*-values are in parentheses. *** means the significance level is 1%; ** means the significance level is 5%; and * means the significance level is 10%.

### 6.2. Heterogeneity Analysis Based on Different Regions

Due to the vast territory of our country, there are significant differences in economy, culture, and environment. The impact of rural road construction on different regions may also be quite different. Therefore, this paper selects the eastern, central, and western regions for regression analysis.

It can be seen from Table 12 that the construction of rural roads has significantly promoted the sustainable development of agriculture in the eastern and central regions, with a more significant impact on the central regions. However, the construction of rural roads has a particular restrictive effect on the sustainable development of agriculture in the western region, but the result is not significant. This paper also carried out a sub-regression for each region to explore the reasons for the spatial heterogeneity and found that: (1) the construction of rural roads in the eastern region has significantly promoted the sustainable development of society and resource systems; (2) the construction of rural roads in the central region has promoted the sustainable development of the population and resource system, but on the other hand, has had a negative impact on the environmental system; and (3) generally speaking, rural road construction has no significant impact on the sustainable agricultural development in the western region. Still, from a sub-item perspective, rural

road construction has significantly promoted the sustainable development of local society, economy, and environment, but restrained the sustainable development population and resources.

**Table 12.** Heterogeneity regression results based on different regions.

| VAR | (1) | | (2) | | (3) | |
|---|---|---|---|---|---|---|
| | **Eastern Region** | | **Central Region** | | **Western Region** | |
| road1 | 0.0436 *** | | 0.1531 *** | | −0.0128 | |
| | (4.3519) | | (3.9929) | | (−1.3310) | |
| road2 | | 0.0333 *** | | 0.1603 *** | | −0.0159 |
| | | (3.0377) | | (3.6768) | | (−1.6110) |
| theil | 0.0036 | 0.0060 | 0.1988 * | 0.1994 * | −0.0003 | 0.0002 |
| | (0.3551) | (0.5695) | (1.9384) | (1.7680) | (−0.0841) | (0.0629) |
| tech | −1.5732 *** | −1.6231 *** | 0.7027 | 0.6038 | −0.0955 | −0.1064 |
| | (−7.4125) | (−7.4658) | (0.6142) | (0.4791) | (−0.9183) | (−1.0256) |
| indus | −0.0338 ** | −0.0275 * | −0.0760 | −0.0787 | 0.0086 | 0.0087 |
| | (−2.0774) | (−1.6492) | (−1.3511) | (−1.2673) | (0.8052) | (0.8094) |
| lnGDP | −0.0878 *** | −0.0833 *** | 0.1196 ** | 0.1148 * | 0.0222 *** | 0.0236 *** |
| | (−6.0631) | (−5.6157) | (2.0850) | (1.8169) | (2.7455) | (2.9180) |
| lnurban | −0.0130 | −0.0077 | 0.5989 | 0.6228 | 0.0021 | 0.0041 |
| | (−0.3315) | (−0.1896) | (1.5761) | (1.4942) | (0.1283) | (0.2509) |
| lnsocialcare | −0.0178 *** | −0.0168 *** | −0.0106 | −0.0043 | −0.0056* | −0.0055 * |
| | (−3.1221) | (−2.8715) | (−0.4431) | (−0.1633) | (−1.9487) | (−1.9196) |
| inexport | −0.0210 ** | −0.0237 ** | 0.0101 | 0.0248 | −0.0095 | −0.0080 |
| | (−2.1216) | (−2.3440) | (0.0686) | (0.1527) | (−0.7708) | (−0.6518) |
| lnfdi | 0.0002 | −0.0003 | 0.0224 * | 0.0197 | 0.0027 *** | 0.0027 *** |
| | (0.0504) | (−0.0961) | (1.6925) | (1.3406) | (3.4526) | (3.4570) |
| financial | 0.1129 | 0.1286 * | 0.3917 | 0.4004 | −0.0154 | −0.0163 |
| | (1.6213) | (1.8059) | (0.9833) | (0.9080) | (−1.2449) | (−1.3197) |
| lneduc | −0.0281 | −0.0611 | −0.1415 | −0.1388 | 0.0078 | 0.0073 |
| | (−0.4144) | (−0.8801) | (−1.0670) | (−0.9489) | (0.5977) | (0.5572) |
| W*X | Y | Y | Y | Y | Y | Y |
| rho | −0.1639 ** | −0.2035 | −0.5220 *** | −0.4182 *** | −0.4492 *** | −0.4458 *** |
| | (−2.1020) | −2.5748 | (−6.5116) | (−4.8511) | (−4.0361) | (−4.0032) |
| R-sq | 0.9789 | 0.9778 | 0.8938 | 0.8889 | 0.9862 | 0.9862 |
| LM-lag | 11.6991 *** | 10.3868 *** | 7.9041 *** | 7.9138 *** | 8.7914 *** | 8.7579 *** |
| R-LM-Lag | 37.9546 *** | 39.8765 *** | 3.3173 * | 3.2783 * | 2.7082 * | 2.7105 * |
| LM-error | 0.4917 | 0.9437 | 10.6227 *** | 10.6181 *** | 7.1396 *** | 7.0987 *** |
| R-LM-error | 26.7473 *** | 30.4335 *** | 6.0359 ** | 5.9826 ** | 1.0564 | 1.0514 |
| Hausman Test | 77.4712 *** | 236.6183 *** | 24.6180 * | 20.1814 * | 49.6034 *** | 21.5360 ** |
| Obs | 204 | 204 | 153 | 153 | 170 | 170 |

Notes: The *t*-values are in parentheses. *** means the significance level is 1%; ** means the significance level is 5%; and * means the significance level is 10%.

In Table 13, the results show that the construction of rural roads has a heterogeneous effect on the sustainable development of agriculture in various regions. Therefore, each region needs to formulate transportation policies suitable for local development according to local conditions. For the eastern and central regions, although the construction of rural roads has significantly promoted the sustainable development of agriculture, its impact on the economy is not significant. Moreover, it has a greater negative impact on the environmental sustainability of the central region. A possible reason is that the better development resources in the central and east regions have attracted a large number of residents to flow in from the western region. That leads to a faster growth rate of the local population than the growth rate of total agricultural output, causing the agricultural output per person to change slowly. Thus, its impact on the economy is not significant. In Figure 6, we could find that from 2002 to 2018, the total agricultural output value of the eastern and central regions has been much higher than that of the western region. The growth rate is also faster than that of the western region. However, its per capita amount is not much

different from that of the western region. Therefore, although rural roads have dramatically promoted the total agricultural output in the eastern and central regions, the per capita amount has not increased much due to large numbers of people. The variable selected in the economic system of this article is the amount per person in each region, which can explain why the construction of rural roads has not significantly promoted the sustainable development of the economic system in the eastern and central regions. In addition, the central region also needs to pay attention to the negative impact of rural road construction on the environment, focusing on reducing fertilization and medicine, soil erosion control, and developing ecological agriculture and high-efficiency agriculture.

**Table 13.** Regional regression results of rural road construction and sustainable development capabilities of population, society, economy, resources, and environment.

| Region | (1) Population | | (2) Social | | (3) Economic | | (4) Resource | | (5) Environment | |
|---|---|---|---|---|---|---|---|---|---|---|
| | road1 | road2 | road1 | road2 | road1 | road2 | road1 | road2 | road1 | road2 |
| Eastern | −0.0004 (−0.8437) | 0.0017 (0.6471) | 0.0181 *** (3.4886) | 0.0049 (0.8621) | 0.0030 (1.2191) | 0.0047 * (1.6508) | 0.0069 *** (5.2096) | 0.0079 *** (5.6844) | −0.0004 (−0.8437) | −0.0007 (−1.4859) |
| Central | 0.0809 *** (3.6005) | 0.0831 *** (3.6000) | 0.0001 (0.0288) | 0.0001 (0.0443) | 0.0007 (0.0945) | 0.0020 (0.2558) | 0.0118 *** (2.7868) | 0.0113 *** (2.5792) | −0.0014 ** (−2.1587) | −0.0015 ** (−2.2584) |
| Western | −0.0019 *** (−2.7714) | −0.0019 *** (−2.7987) | 0.0124 *** (3.6803) | 0.0134 *** (3.8856) | 0.0084 ** (2.3879) | 0.0086 ** (2.3761) | −0.0183 *** (−3.8188) | −0.0206 *** (−4.2331) | 0.0032 *** (2.6653) | 0.0033 *** (2.6056) |

Notes: The *t*-values are in parentheses. *** means the significance level is 1%; ** means the significance level is 5%; and * means the significance level is 10%.

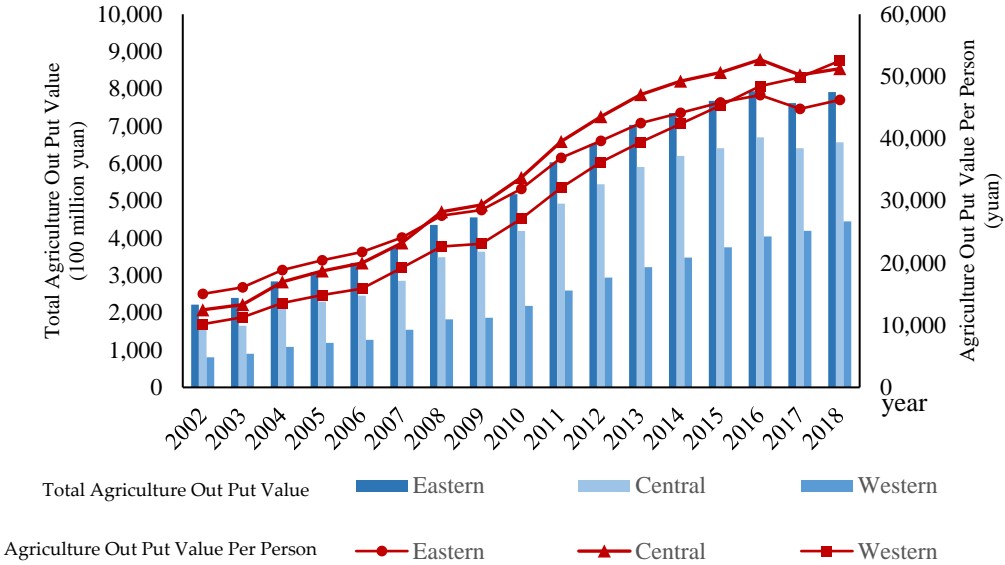

**Figure 6.** The total agricultural output value and per capita agricultural output value in the eastern, central, and western regions.

## 7. Conclusions and Policy Implications

This paper uses China's 2002–2018 provincial panel data to select indicators from five dimensions of population, society, economy, resources, and environment based on the characteristics of sustainable agricultural development. It calculates the sustainable agricultural development index of China's provinces through the entropy method. Then, we use these variables to build a spatial measurement model for exploring the impact of rural road construction on the sustainable development of regional agriculture in China. We found that:

(1) On the whole, the promotion of rural road construction to the sustainable development of agriculture is greater than the restraining effect, which is conducive to the sustainable development of agriculture. Rural road construction reduces the cost of pop-

ulation, capital, and commodity transportation, promotes the transfer of rural surplus labor, and improves agricultural mechanization and fixed assets, thereby enhancing the sustainability of population, society, and resources. However, the reduced transportation costs also bring down the prices of chemical fertilizers and pesticides. The excessive use of these chemicals by farmers has caused environmental problems such as soil erosion and land acidification, thereby inhibiting the sustainable development of the agricultural environment.

(2) In terms of time effect, the impact of rural road construction on the sustainable development of agriculture in 2008–2018 was much more significant than that in 2002–2007, indicating that the large-scale investment in transportation infrastructure in 2008 had a more significant impact on the sustainable development of agriculture. At the same time, from the regression results, the impact of rural road construction on the sustainable development of China's agriculture has a lagging effect more significant than the current effect. Meanwhile, the two lagging phases' results are lower than the lagging one phases' effect. It can be seen from the results that the impact of rural road construction has a lag in time, and at the same time, the impact has the characteristic of diminishing margins.

(3) From a regional perspective, the impact of rural road construction on the sustainable development of agriculture in different regions of China is different. The construction of rural roads has significantly promoted the sustainable development of agriculture in the eastern and central regions. However, its impact on the sustainable economic development of the two places is not significant, and it also has a substantial negative impact on the environmental sustainability of the central region. For the western region, the overall effect of rural road construction on sustainable agricultural development is not significant. However, we find that the construction of rural roads has significantly reduced the sustainable development of population and resources in the area and has improved the sustainable development effects of society, economy, and environment. From the results, the siphoning effect brought by the construction of rural roads has caused a considerable loss of talents and capital in the western region, resulting in a decline in the sustainability of its population and resource systems. This effect is offset by the positive social, economic, and environmental effects.

The above research conclusions have the following policy implications for China's agricultural development:

Firstly, the Chinese government needs to continue to improve the construction of rural road networks to promote sustainable agricultural development. Overall, although the construction of rural roads harms the environment, its positive effects on population, society, resources, etc., can make up for it. At this stage, China should continue to improve the construction of rural roads and promote increased agricultural labor productivity to ensure the overall production capacity of major agricultural products. Then, we should protect agricultural resources and the ecological environment while grains are steadily improved to achieve a stable production development and an ecologically friendly situation.

Secondly, it is necessary to conduct a long-term evaluation from the whole life cycle perspective to save costs to the greatest extent, improve the quality of the road, and increase the ultimate social benefits. Rural road construction has a certain lag in the sustainable development of agriculture. In the short term, if only emphasizing the speed and scale of road construction and ignoring environmental protection, it will not be conducive to the sustainable development of agriculture in the long run.

Finally, local governments need to adopt measures to local conditions and give full play to the positive effects of rural road construction on the sustainable development of local agriculture. At the same time, they should also focus on its negative externalities to maximize the positive impact of rural road construction on the local area. For the eastern and central regions, governments should guide local farmers in the scientific application of chemical fertilizers and pesticides, increase the application of organic fertilizers, and implement straw returning to the field for reducing the impact of chemical fertilizers and pesticides pollution on agricultural production. For the western region, the government

needs to create preferential policies, such as talent subsidies and agricultural subsidies, to attract talent and capital inflows. Accumulated local human capital and the development of agricultural mechanization will be beneficial for increasing their agricultural productivity and labor productivity in the western region.

In addition, the research conclusions of this article can provide a specific reference for other regions and countries that focus on agricultural development while vigorously developing rural roads. For example, areas with relatively backward development need to pay attention to the siphoning effect of the construction of rural roads. The government needs to increase subsidies to the education and agricultural departments to maintain local talents and capital and promote the improvement of the local agricultural population and resource sustainability. As for developed areas, more attention needs to be paid to the impact of rural road construction on the environmental sustainability of agriculture.

**Author Contributions:** Conceptualization, Z.Z. and J.D.; methodology, W.L.; software, Z.Z.; validation, Z.Z., J.D. and S.G.; formal analysis, S.G.; investigation, Z.Z.; resources, Z.Z.; data curation, S.G.; writing—original draft preparation, Z.Z.; writing—review and editing, J.D.; visualization, S.G.; supervision, W.L.; project administration, W.L.; funding acquisition, Z.Z., J.D. and W.L. All authors have read and agreed to the published version of the manuscript.

**Funding:** This research was funded by the National Social Science Foundation research project (Grant No. 21AZD019) and the Fundamental Research Funds for the Central Universities (Grant No. 2021YJS068), China.

**Institutional Review Board Statement:** Not applicable.

**Informed Consent Statement:** Not applicable.

**Data Availability Statement:** The data that support the findings of this study are available in "China Statistical Yearbook", provincial and municipal statistical yearbooks. These data were derived from the following resources available in the public domain: http://www.stats.gov.cn/tjsj/ndsj/ (accessed on 27 September 2021).

**Conflicts of Interest:** The authors declare no conflict of interest.

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
