# Peer review of "Can Rural Road Construction Promote the Sustainable Development of Regional Agriculture in China?"

_sustainability, doi:10.3390/su131910882_

Round 1

Reviewer 1 Report

Review of the paper titled „Can Rural Roads Construction Promote the Sustainable Development of Regional Agriculture in China? Under consideration in the Sustainability journal:

  • The introduction is too wordy for me. It presents a lot of information about road grid, but once reading it one may be lost in the objective, novelty of the paper, its contribution, etc. One may want to see a brief summary of the main findings and policy implications at the end. Consider shifting some part of the text into stylized facts section (with information on road network, China specific situation).
  • Consider rephrasing Hypothesis 5. The impact of rural road construction on the sustainable development of agriculture is lagging. -> There is a [certain] lag in the impact of rural road construction on the sustainable development of agriculture
  • Consider rephrasing Hypothesis 6. The impact of rural road construction on the sustainable development of agriculture in different regions is heterogeneity. -> heterogeneous?
  • What with the effect of other roads’ types on regions, which can be omitted in the paper? Wouldn’t it be more desirable to take into account different types of roads simultaneously? I suspect there are different kinds of roads in particular regions of China.
  • This part is ambiguous and needs rewriting to the point: (page 13) “paper finds that the mileage of provincial highways graded according to the engineering grade is published in the Yearbook. Still, there is no administrative grade data (national highway, provincial high way, county road, and township road). The mileage of highways graded by the engineering grade provides a breakthrough for empirical measurement. By consulting the data and using the indicators of Liu [56], Jiang [57], we also find that the areas connected by rural roads classified according to engineering and technical grades have significant differences in economic and administrative status. Compared with highways and first-class roads, second-class roads have a higher probability of connecting suburban or rural areas across regions while connecting major cities. Thus, they are included in rural roads construction indicators that could avoid measurement errors in regression. At present, it has been studied that the per capita mileage of four-class roads is used as the proxy for rural roads construction. But this method is not entirely reasonable. The agricultural economy mainly dominates county towns.” Also next paragraph needs some polishing. From reading these paragraphs I would like to have a clear distinction what types of roads have you chosen as the dependent variable, similarly the reason for exclusion particular types. Maybe a table with particular roads’ types would enhance the clearness of this part of manuscript? Second problem with these two first paragraphs of subchapter 3.3.2 is that the reader doesn’t know whether you write about China or in general. Take into consideration that international reader does not have to know the specific organisation of road network/regional geography/administrative division of China.
  • One could try to find explanation for the lack of significance in Table 3 from 2015.
  • The analysis presented in the study seems to be to big extent rely on the case of China and its problems. Authors should to a larger extent try to internationalise the problem and formulated policy recommendations.
  • Add a reference in paragraph starting from line no. 282 (page 7). https://doi.org/10.1007/s00168-019-00947-6

Minor issues:

  • (page 9): “To verify the above assumptions, we will build the following econometric model for empirical analysis” -> we build ..
  • Check the alignment of values in cell of Table 3, as it might be misleading.

Reviewer 2 Report

· The research aim and object are neither indicated in the Abstract nor in the Introduction – they must appear.

· It is mentioned in the text that “Therefore, the construction of rural road infrastructure has always attracted China’s government’s attention” – what documents (if they are) explore the analyzed problems at the regional level?

· The authors distinguish population sustainability from social sustainability? Why is it important? Usually population is the primary indicator of social dimension. Such an approach is debatable. 

· Looking at Figure 1 - not really clear where the beginning is.

· It is suggested at the beginning of “Methods and Data” to give a logical research scheme – to explore the steps of how the research was made and then to describe it. Also, there could be some references to other research where the used econometric model was applied in other countries.

· How does the “Conclusions and Policy Implications” reflect the hypothesis which were given in the text? Maybe it is possible to give more detailed conclusions.
